# Automated Over-the-Top Service Copyright Distribution Management System Using the Open Digital Rights Language

Wooyoung Son [1], Soonhong Kwon [2], Sungheun Oh [3] and Jong-Hyouk Lee [2,*]

[1] Protocol Engineering Laboratory, Sejong University, Seoul 143-747, Republic of Korea; wooyoung@pel.sejong.ac.kr
[2] Department of Computer and Information Security and Convergence Engineering for Intelligent Drone, Sejong University, Seoul 143-747, Republic of Korea; soonhong@pel.sejong.ac.kr
[3] Technical Research Center, DigiCAP, Seoul 121-904 , Republic of Korea; shoh@digicaps.com
[*] Correspondence: jonghyouk@sejong.ac.kr

**Abstract:** As the demand and diversity of digital content increase, consumers now have simple and easy access to digital content through Over-the-Top (OTT) services. However, the rights of copyright holders remain unsecured due to issues with illegal copying and distribution of digital content, along with unclear practices in copyright royalty settlements and distributions. In response, this paper proposes an automated OTT service copyright distribution management system using the Open Digital Rights Language (ODRL) to safeguard the rights of copyright holders in the OTT service field. The proposed system ensures that the rights to exercise copyright transactions and agreements, such as trading of copyright, can only be carried out when all copyright holders of a single digital content agree based on the Threshold Schnorr Digital Signature. This approach takes into account multiple joint copyright holders, thereby safeguarding their rights. Furthermore, it ensures fair and transparent distribution of copyright royalties based on the ratio information outlined in ODRL. From the user's perspective, the system not only provides services proactively based on the rights information specified in ODRL, but also employs zero-knowledge proof technology to handle sensitive information in OTT service copyright distribution, thereby addressing existing privacy concerns. This approach not only considers joint copyright holders, but also demonstrates its effectiveness in resolving prevalent issues in current OTT services, such as illegal digital content replication and distribution, and the unfair settlement and distribution of copyright royalties. Applying this proposed system to the existing OTT services and digital content market is expected to lead to the revitalization of the digital content trading market and the establishment of an OTT service environment that guarantees both vitality and reliability.

**Keywords:** Hyperledger Fabric; ODRL; chaincode; OTT service; Threshold Signature Scheme; copyright distribution management systems



## 1. Introduction

Since the emergence of COVID-19, the demand for digital content has surged, invigorating the digital content market. Nonetheless, in contrast to the positive developments associated with this growth, the central stakeholders of the digital content industry, particularly the copyright holders, find themselves in a predicament where their legitimate rights are not being sufficiently safeguarded.

Spotify, a major music streaming platform, has introduced initiatives to protect the rights of music copyright holders, particularly against illegal reproduction and distribution, including stream ripping. However, these copyright holders, who use Spotify's service, have highlighted issues with the method of royalty settlement and distribution. The root of these problems is the 'black box' issue, where music ownership is split among different groups, resulting in delays or non-payment of the rightful royalties to the copyright holders [1,2]. The

same problem is evident in Theme/Background/Signal (TBS) music as well. In addition, the video content copyright sector is facing challenges related to copyright infringement caused by widespread account sharing, and issues arising from the lack of transparency in the settlement and distribution of royalties [3,4].

Recently in the OTT service industry, there has been a rise in illegal replication and distribution of digital video content. This is driven by practices like stream ripping, rampant password sharing among users, credential fraud, deceptive consumer endpoints, and man-in-the-middle attacks on Content Delivery Networks (CDNs). As a result, the rights that should be protected for copyright holders are increasingly being violated [5].

Furthermore, when copyright holders delegate the management of their digital content rights to agents, the opaque structure of this arrangement frequently results in their inability to clearly access information about the usage history and the status of royalty settlements and distributions for their digital content.

In order to protect the rights of copyright holders, there have been significant research and development efforts focused on creating a fair and transparent copyright management system. This involves the use of technologies like Digital Rights Management (DRM) and Blockchain.

However, a majority of these studies fail to offer dedicated solutions for safeguarding sensitive data involved in the management and distribution of digital content copyrights and in royalty settlements. This lack of protective measures poses a risk of privacy breaches and, in cases of data leaks, could lead to disputes between companies or between companies and individuals.

Digital content creation can involve just one copyright holder, but it often includes several joint copyright holders. Yet, most studies fail to investigate methods for managing royalty settlements and distributions in cases involving multiple joint copyright holders, which limits the practical application of these solutions in the industry. Moreover, while current policies and regulations demand the consent of all copyright holders for transactions involving digital content, this critical requirement is also overlooked in the study.

Consequently, the ODRL-based automated OTT service copyright management system proposed in this study not only suggests a method for settling and distributing royalties with consideration for multiple stakeholders, specifically joint copyright holders, but also provides a mechanism for these holders to express their consent effectively and logically through signatures in the event of a copyright transaction. Moreover, this system addresses the sensitive data involved in digital content distribution and royalty settlements by proposing privacy protection measures. This approach overcomes the shortcomings and limitations that current research has not yet resolved, aiming to offer a viable system for the digital content industry.

The layout of this paper is outlined as follows. In Section 2, the paper delves into the technical concepts of Hyperledger Fabric, smart contracts, and ODRL. Section 3 provides an analysis of the current Blockchain-based methods for copyright management, including royalty settlement and distribution. Section 4 elaborates on the ODRL-based automated OTT service copyright management system proposed in this study. Section 5 explains the key modules of this system. Performance analysis is conducted in Section 6 to showcase the system's efficiency, while Section 7 explores the technical approaches and constraints associated with the proposed system. The paper is wrapped up in Section 8.

## 2. Background

### 2.1. Hyperledger Fabric

Hyperledger Fabric is a Blockchain platform designed for enterprise use. Before version 2.0, it had no concept of separate cryptocurrency and managed the nodes included in each organization independently. Since Hyperledger Fabric is an enterprise Blockchain, it employs an Membership Service Provider (MSP) to perform identity verification of nodes to participate in the Blockchain network, enabling access control and establishing accountability in case of issues. Additionally, when composing chaincode, it allows configurations

based on general programming languages, ensuring versatility. It also guarantees confidentiality and privacy by allowing only the authorized peers participating in the channel to access the data [6–10].

As can be seen in Figure 1, various components operate organically on the Hyperledger Fabric network [6]. Table 1 below shows the main roles of components in the Hyperledger Fabric architecture [6–10].

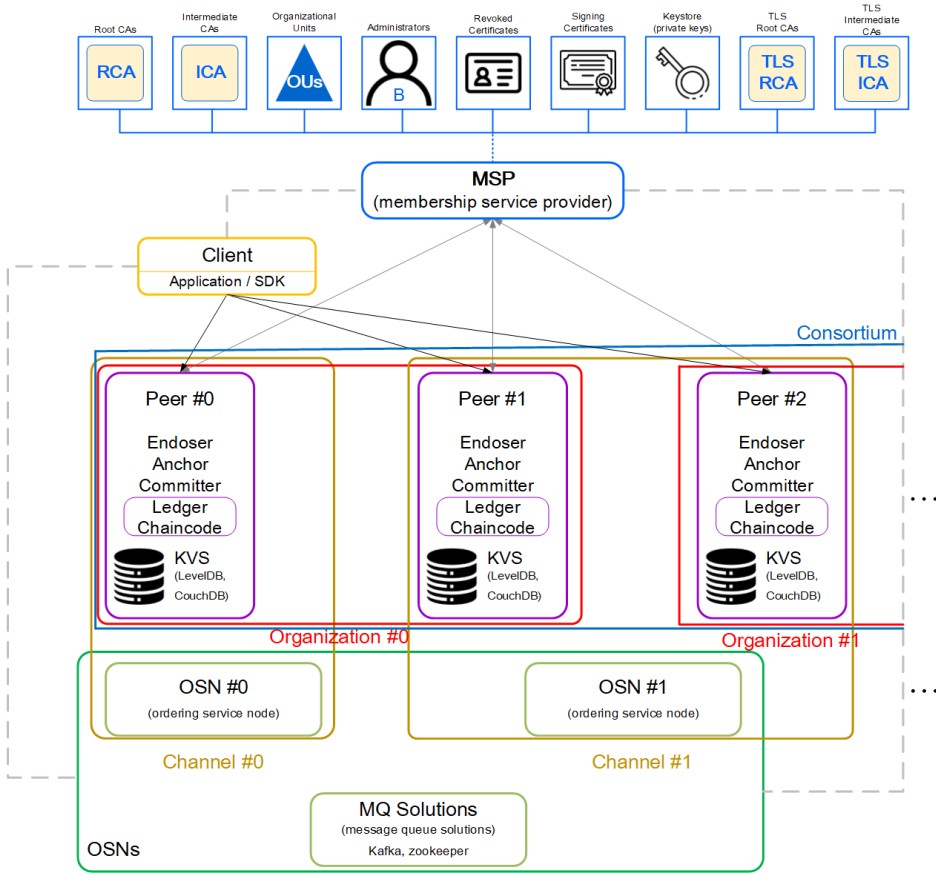

**Figure 1.** Architecture of Hyperledger Fabric.

In the case of transaction processing and recording in the ledger based on the components in Figure 1, operations between components are performed as shown in Figure 2.

1. The client forwards the signed transaction proposal to the predefined Endorser.
2. The Endorser checks whether the signature of the transaction proposal received from the client is valid, and if valid, executes the chaincode based on the parameters of the transaction proposal and delivers the transaction results (including Read set and Write set) to the client.
3. The client checks whether the signature value of the returned transaction result matches the Endorser's signature value.
4. If the reliability of the transaction result returned through Process 3 is guaranteed, it is delivered to the ordering service node.
5. As the ordering service node receives already verified transaction results from the client, it organizes them by channel and time to create a transaction block per channel.
6. Transaction blocks per channel generated by the ordering service node are broadcast to all peer nodes on the Blockchain network.
7. All peer nodes verify the validity by checking the contents of the R/W set, such as guarantee policy and ledger state changes, based on the received transaction block, and if verification is successful, they commit it to the world state.

**Table 1.** Hyperledger Fabric architecture key Component role.

| Component | Description |
|---|---|
| Channel | Components that establish and manage group permissions for transactions on a Blockchain network by group |
| Organization | Components that manage permissions to peer nodes by organization within a Blockchain network as well as access to network participants |
| Peer Node | Components that process transactions on a Blockchain network, manage and store ledger and chaincode (Maintain Blockchain network) |
| Ordering Service Node | Components that contain information about channels within the Blockchain network and act as administrators |
| Membership Service Provider (MSP) | Components that perform roles related to identity authentication in the Blockchain network to achieve access control and manage the roles and privileges of each component |
| Certificate Authority (CA) | Components that act as a certification authority in the Blockchain network, providing the information required for encryption authentication in MSP to achieve access control and manage the roles and privileges of each component. |

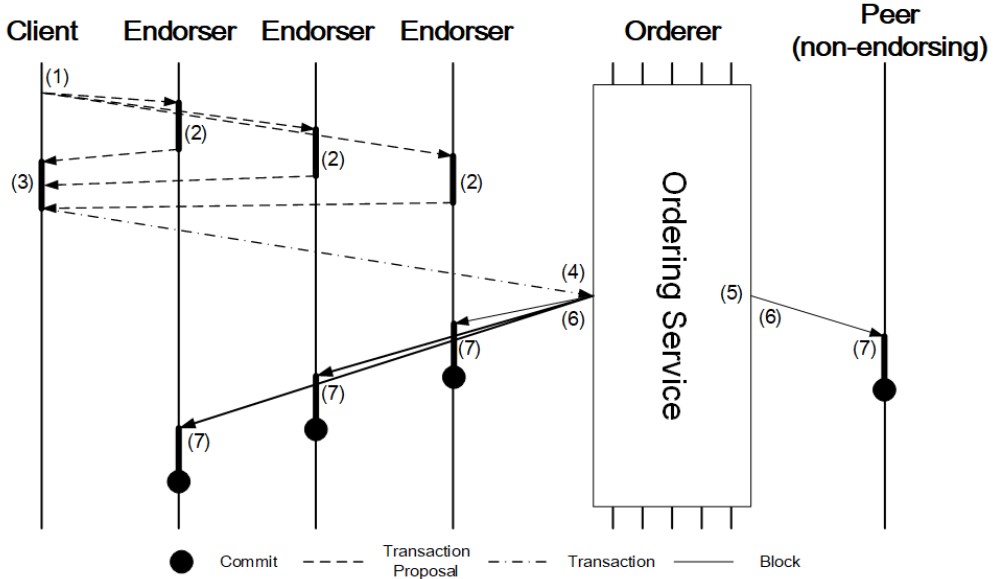

**Figure 2.** Transaction flows in Hyperledger Fabric network.

Hyperledger Fabric is currently being employed in areas such as smart homes, healthcare, education, and digital content, and the trend shows that its application fields are gradually increasing [11–14].

*2.2. Smart Contract*

Blockchain technology is typically divided into public Blockchains like Bitcoin and private Blockchains like Hyperledger Fabric. As illustrated in Figure 3, such Blockchains are generally composed of six layers. This ranges from the 'Data Layer', which sets up the data structure including data blocks, timestamps, and other data, to the 'Application Layer', which enables the development of various application services based on Blockchain [15].

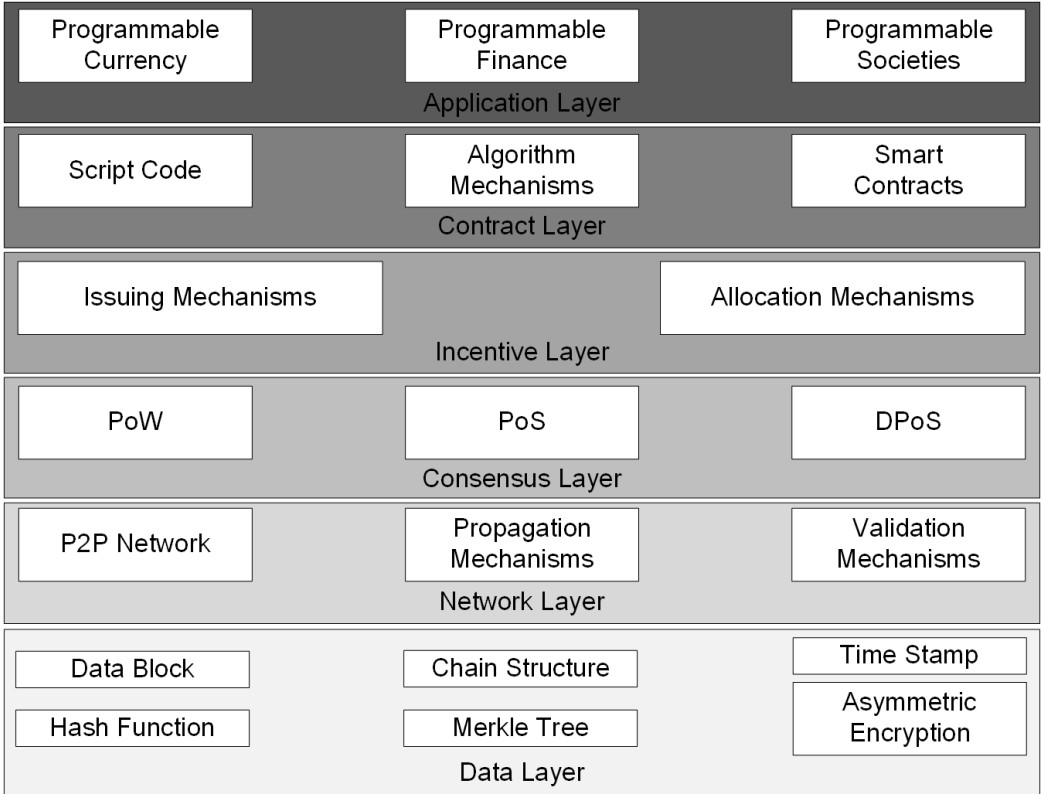

**Figure 3.** The structure of Blockchain layers.

Within these layers of Blockchain technology, the 'Contract Layer' plays a pivotal role in configuring automated events in the Blockchain network. This layer, whether it is Ethereum's smart contract or Hyperledger Fabric's chaincode, is programmed to meet specific contract conditions, enabling the execution of user-intended events in an automated manner [15].

Smart contract development typically utilizes programming languages like Solidity and Vyper, and leverages Integrated Development Environments (IDEs) such as Remix and Truffle. For interfacing with nodes, it employs JSON-RPC. The process of creating a smart contract involves writing code, converting it into bytecode, and then distributing this bytecode across the nodes in the Blockchain network. Furthermore, a smart contract has a lifecycle that encompasses the process of accessing a deployed smart contract on the Blockchain network through calls [15–18].

In contrast, chaincode supports general-purpose programming languages such as GO and JavaScript. It is developed to meet specific user requirements using IDEs like Chaincoder and Goland [19–21]. For node interface communication, the gRPC method is employed. Typically, chaincode is written, compiled into an executable file, and then packaged. This packaged chaincode is then distributed to Endorser nodes participating in a channel. Accessing chaincode involves a calling process similar to that of smart contract. However, chaincode differs in that it executes by requesting a proposal, and a transaction is submitted to the Orderer only upon receiving a return value. Blocks created through this process are broadcasted to nodes within the channel via the Ordering Service. Each peer node receiving these blocks adds them and commits them to the world state, ensuring a coherent and secure transaction process.

Smart contract, utilized in public Blockchains, is distributed across all nodes, leading to a distinct approach in access control compared to chaincode. On the other hand, chaincode is only distributed to authorized nodes within a specific channel. This difference significantly impacts how access is controlled. Furthermore, the immutable nature of Ethereum, a public Blockchain, prohibits data updates, rendering smart contract updates infeasible.

In contrast, Hyperledger Fabric, a private Blockchain, allows data updates and deletions, contingent on governance permissions. This flexibility enables updates in Hyperledger Fabric that are not possible in Ethereum. Table 2 provides a comparative overview of the key characteristics of smart contract and chaincode [22].

**Table 2.** Comparison of smart contract and chaincode.

| Item | Smart Contract | Chaincode |
|---|---|---|
| Language | Solidity, Vyper | Go, JavaScript |
| IDE | Remix, Truffle | Chaincoder, Goland |
| Node Communication Interface | JSON-RPC | gRPC |
| Access Control | Not Supported | Supported |
| Deployment Policy | Deployed to all nodes within the Blockchain network | Deployed only to authorized nodes within the channel |
| Availability of Updates | Not Supported | Supported |
| Lifecycle | 1. Generation process 2. Deployment process 3. Calling process | 1. Generation and packaging process 2. Deployment process 3. Calling process |

*2.3. ODRL*

With the increasing diversity of digital content and the widespread dissemination of numerous digital content, the need for a standardized management of digital rights and permissions to protect copyrights has grown. The ODRL emerged in this context, aiming to address the increasingly complex and extensive digital rights management. It allows copyright holders to precisely specify the restrictions they wish to impose on their digital content, and based on this, end-users can clearly understand the usage rights of the digital content, thereby protecting the rights of the copyright holders. The ODRL is designed to be scalable and flexible, enabling the expression of complex rights and permissions through various Classes. The main Classes of the ODRL are as shown in Table 3 [23].

The ODRL specifies digital asset rights in detail through these Classes. A typical ODRL Policy allows the 'Assigner' to provide the 'Asset' to the 'Assignee' via 'Rule', which can introduce more granular constraints through 'Action' and 'Constraint'. Such a structured framework ensures that rights are precisely defined and applied, enabling effective management of restrictions like a copyright transfer and access limitations of digital content.

Listing 1 defines rights and restrictions related to the use of digital content based on the ODRL [23]. It indicates an agreement or contract between parties through the 'Agreement' subclass of the 'Policy' Class. This 'Policy' grants rights to the 'Asset' set as 'http://example.com/DigitalContent:0709', conferred by 'http://example.com/Agent1', designated as 'assigner'. The rights are assigned to 'http://example.com/Usergoup1', designated as 'assignee'. Through 'PartyCollection', it is understood that the rights are granted to a group, not just an individual. 'PartyCollection' allows specifying that only those meeting certain criteria, defined in 'refinement', receive rights. The 'dateTime' operand on the left signifies date and time, while 'lteq' represents 'less than or equal to', meaning the left operand's value must be less or equal to the right operand, which represents the service expiry date. Thus, through the 'refinement' condition, it is discerned that 'Usergroup1' is a collection of 'assignees' with usage rights until '2024-07-09'. The granted rights include the 'play' action within the 'Action' Class, permitting playback of the target. The constraint on 'Action' includes 'schema:age' as the left operand, and 'gteq', meaning 'greater than or equal to'. The right operand being '19' indicates that the digital content playback right is granted only if the user's age satisfies the condition of being 19 years or older. Therefore,

through Listing 1's ODRL, users in 'UserGroup1', grouped by the service expiry date, can play 'http://example.com/DigitalContent:0709' only if they are at least 19 years old.

**Table 3.** Main classes of the ODRL.

| Class | Description |
|---|---|
| Policy | This Class forms the fundamental structure of ODRL and is composed of subclasses such as 'Set', 'Offer', and 'Agreement'. It defines the 'Action' of the 'Party' concerning a specific 'Asset', thereby clearly articulating 'Permission', 'Prohibition', and 'Duty'. |
| Asset | This Class represents the content and services that are the subjects of the rules, such as digital books, music, and streaming videos. The subclass 'AssetCollection', which groups together multiple assets, is particularly useful for uniformly applying the same rights expressions to numerous contents or services |
| Party | This Class represents entities that participate in and perform specific roles within the rules, which can include groups, individuals, or organizations. The 'Party' can be categorized into 'Assigner', who establishes and proposes policies, and 'Assignee', who accepts these policies. Additionally, it includes the subclass 'PartyCollection', which is used for grouping and representing multiple entities. |
| Action | This Class represents actions that the 'Party' can perform on an 'Asset', such as streaming and downloading. |
| Constraint | This Class represents conditions and restrictions applied to the subclasses of 'Rule', namely 'Permission', 'Prohibition', and 'Duty', thereby providing a more detailed explanation of the scope and application of authority. |
| Rule | This Class, through its subclasses 'Permission', 'Prohibition', and 'Duty', restricts the actions of 'Party' regarding the 'Asset'. |

**Listing 1.** ODRL Information Model V2.2.

```
 1  {
 2      "@context": "http://www.w3.org/ns/odrl.jsonld",
 3      "@type": "Agreement",
 4      "uid": "http://example.com/policy:9090",
 5      "profile": "http://example.com/odrl:profile:07"
 6      "permission": [
 7          {
 8              "target": {
 9                  "uid": "http://example.com/DigitalContent:0709"
10              },
11              "assigner": "http://example.com/Agent1",
12              "assignee": {
13                  "@type": "PartyCollection",
14                  "source": "http://example.com/Usergroup1",
15                  "refinement": [
16                      {
17                          "leftOperand": "dataTime",
18                          "operator": "lteq",
19                          "rightOperand": { "@value": "2024-07-09
                                ", "@type": "xsd:data" }
20                      }
21      ]
```

```
22              },
23              "action": "play",
24              "constraint": [
25                  {
26                      "leftOperand": "schema:age",
27                      "operator": "gteq",
28                      "rightOperand": { "@value": "19", "@type": "xsd
                            :integer" }
29                  }
30              ]
31          }
32  ]
33  }
34  }
```

In today's digital environment, where digital content is continuously shared and consumed across borders and platforms, the need for a standardized language of rights is more crucial than ever. In this context, the ODRL has become a key indicator, offering a consistent approach to the representation and application of digital rights.

ODRL, used for expressing digital rights, permissions, and obligations, is widely utilized in various fields beyond the digital environment due to its standardized format.

Firstly, ODRL is being employed in the mobility data space as a means to protect user privacy. Mobility data encompasses all data generated by modes of transport and services, as well as by drivers, and while it offers efficiency to users, it also poses privacy risks [24]. To address this, research is underway to protect sensitive mobility data and to safely exchange it by establishing and formalizing policies through the standardized language of ODRL [25].

Additionally, fashion brands delivering products to customers via the internet, in compliance with the General Data Protection Regulation (GDPR) since May 2018, are looking to use ODRL to build an automated compliance check approach. This aims to overcome the limitations of privacy policies applied in formats unrecognized by computers, by automating the search and monitoring of policies applied through ODRL, thereby more effectively protecting customer privacy in the digital transformation of fashion brands [26].

In an era where personal data protection is a major topic of discussion, ODRL is being actively used in a multitude of research projects aimed at privacy preservation. Furthermore, it is being extensively researched and developed across a range of different domains, emphasizing its vital role in the ongoing discourse on privacy and data protection.

## 3. Preliminaries

Since COVID-19, there has been an increase in the consumption of digital content, leading to a more active digital content market. However, with this growth, illegal distributors are replicating and distributing digital content unlawfully, infringing on the rights that should be protected for copyright holders.

As a result, extensive study and development are being conducted to ensure the rights of copyright holders, focusing not only on means of protection, but also on strategies to safeguard sensitive data related to digital content copyrights. Furthermore, continuous monitoring is necessary to detect illegal replication and distribution of digital content. Given the limitations of the human effort in such activities, there is a growing need for automated and proactive copyright management systems. Table 4 presents the copyright distribution management systems that have been researched and developed so far to address these requirements. Table 4, the symbol 'O' indicates that the system proposed in the paper provides the respective functionality, 'X' denotes the absence of that functionality, and '△' signifies that only a part of the functionality is offered.

In this regard, A. Kim and M. Kim [1] proposed a Blockchain and smart contract-based music content distribution management system. This system allows for transparent verification of transaction information and copyright data related to music content. It is

built on the immutable and transparent characteristics of Blockchain, automating a series of processes based on smart contract. Additionally, they presented a model for representing music content, thereby proposing a way to manage large volumes of music content.

**Table 4.** Study trends on Blockchain-based digital content distribution management methods.

| Ref | Copyright Distribution Management Method | Privacy Protection Method | Royalty Settlement and Distribution Method | Smart Contract Signing Method | User Permission Check Method |
|---|---|---|---|---|---|
| [1] | O | X | X | X | X |
| [27] | O | X | O | X | X |
| [28] | O | X | O | X | X |
| [29] | O | X | X | X | X |
| [30] | O | △ | X | X | X |
| [31] | O | X | O | X | X |
| [4] | O | X | O | X | X |
| [3] | O | O | X | X | X |

C.-J. Choi et al. [27] noted the limitations in the settlement and distribution of digital content royalties in the current digital content distribution environment. They aimed to improve these limitations by utilizing Blockchain technology. Additionally, they discussed the challenges in proactive integration between Blockchain networks and external systems. To address this, they proposed a Blockchain-based native currency transfer system with scheduled transfer transactions, ensuring reliability. This system, using a single transaction for automated services, overcomes performance limitations that most Blockchain-based royalty settlement and distribution systems could not solve, thus constituting an efficient system.

Zheng, Xinyu et al. [28] observed the rapid growth of the short film industry, alongside the continuous occurrence of copyright infringement issues. In response, they proposed a Blockchain NFT-based short film copyright transaction system. In this system, when a copyright transaction occurs, the transfer of copyright takes place transparently based on account information in the Blockchain environment. Furthermore, the system facilitates the settlement and distribution of royalties within the Blockchain context. A distinctive feature of this system is its rights protection functionality; if users identify copyright infringement cases, they can collect evidence through a dedicated application. For the first reporter, incentives are issued in the form of tokens, setting this system apart from others.

Varaprasada Rao, K. et al. [29] note that with the digitization of content, illegal replication and distribution have become more effortless, leading to a rapid increase in copyright infringement issues. They also highlighted the limitations of current digital content copyright management models, proposing a Blockchain-based copyright protection system to address these shortcomings. This system allows for the registration of digital content information on the Blockchain via smart contract, ensuring smooth transfer of copyrights during transactions.

Li, Jun et al. [30] analyzed that with the spread of digital content, from a copyright holder's perspective, there is a need for platforms to upload and host digital content online, and from a consumer's perspective, platforms are needed to pay for and download digital content online. Consequently, they developed and introduced LBRY, a Blockchain-based decentralized digital content market. This system anonymizes the identity of copyright holders who publish digital content, thereby preventing the disclosure of the actual identity and protecting against privacy infringement issues. This feature marks a significant distinction from other study and development trends.

Y. Kim et al. [31] discussed the limitations in the current Theme/Background/Signal (TBS) music domain. Firstly, they noted the difficulty in tracking rights changes due to the frequent alterations inherent in TBS music. Secondly, they identified the risk of unfair and unclear royalty settlements and distributions due to complex licensing methods (e.g., compensation payment methods, usage fee payment methods, comprehensive settlement methods, volumetric settlement methods) and centralized royalty management.

To address these issues, they proposed a Blockchain-based TBS music distribution system aimed at improving the unreasonable royalty settlement and distribution practices of the existing TBS music system. While traditional settlement and distribution relied on pre-agreed amounts, proving to be unfair, their proposed system generates usage records through monitoring [4] and bases royalty distributions on the frequency of use. However, as usage records generated through monitoring could contain sensitive information not meant for public disclosure, they utilized selective disclosure technology to ensure that only authorized stakeholders could access sensitive information while keeping it concealed from general users [3,32].

As evident from Table 4, although most studies provide copyright management functionalities, they primarily exhibit limitations in the following four aspects.

(1) Lack of privacy protection measures for sensitive data related to digital content copyright (e.g., information on copyright holders, sales proceeds, etc.).
(2) Absence of smart contract signing approach that considers multiple stakeholders.
(3) Lack of a fair and transparent mechanism for the settlement and distribution of digital content royalties.
(4) Absence of a method to verify users' rights to access digital content.

Digital content copyright data can include sensitive information such as the copyright holder's details, usage fees, or sales/contract information of digital content. Since the system operates on the Blockchain, all nodes participating in the Blockchain network, even those who are not actual stakeholders, can access this sensitive data. This poses a risk of privacy infringement, and with the growing focus on data privacy, it is imperative to explore measures to protect such information.

For digital content copyright transactions to occur, consent from all copyright holders is required. Therefore, for the management of settlement and distribution of royalties via smart contract, it is essential to consider multiple stakeholders, especially in cases of joint copyright holders. Correspondingly, an approach for smart contract signing must be explored. However, most studies have focused only on a single copyright holder and have not separately addressed smart contract solutions for multiple stakeholders, which is a limitation.

One of the biggest issues in the digital content field is the unfair and unclear settlement and distribution of copyright royalties. Currently, royalties are settled and distributed based on the set prices of specific digital content, with most being collected and distributed by trust management organizations. Consequently, copyright holders are unable to verify the exact amount of royalties due for their works. Therefore, a fair and transparent mechanism for royalty settlement and distribution is essential. However, some studies still fail to consider this issue.

Furthermore, from the perspective of digital content distribution management, it is crucial to have a method that proactively verifies a user's rights to access the digital content. However, most studies proposing digital content distribution management solutions tend to omit the process of verifying user access rights, presenting a limitation in their applicability to real-world industrial environments.

## 4. Proposed System

This section outlines the automated OTT service copyright distribution management system utilizing the ODRL. The existing system demonstrated a significant drawback: it overlooked the presence of multiple joint copyright holders in a single piece of digital content and did not adequately address the settlement and distribution of copyright royalties or the proactive provision of rights from the consumer perspective. Moreover, the distribution of OTT service copyrights, compounded by the reliance on Blockchain technology, raised ongoing privacy concerns due to the lack of protective measures for sensitive information. The proposed system addresses these issues by enabling multiple joint copyright holders of a single digital content piece to actively engage in OTT service distribution through direct signatures, thereby ensuring a logical and equitable distribution

process. Copyright royalties are allocated fairly and transparently, based on the ratios outlined in the ODRL. Furthermore, the system enhances consumer convenience by proactively offering services based on digital content usage rights as specified in the ODRL. To safeguard privacy, particularly concerning sensitive data like social security numbers, the system employs zero-knowledge proof range proof techniques, verifying individuals as over 19 years old without exposing private information [33].

Accordingly, the main goals and contributions of the system proposed in this paper can be as follows.

(1) The proposed system employs zero-knowledge proofs to safeguard sensitive information in digital content transactions, effectively preventing privacy infringements.

(2) The proposed system ensures the distribution of digital content is based on the signatures of all joint copyright holders, taking into account multiple joint copyright holders.

(3) The proposed system automatically verifies the usage rights of consumers as stated in the ODRL and provides digital content accordingly.

(4) The proposed system automatically extracts information related to copyright royalties distribution from the ODRL and performs settlement and distribution of copyright royalties, considering the joint copyright holders.

To achieve these objectives, this paper proposes the automated OTT service copyright distribution management system utilizing the ODRL, with the system architecture as illustrated in Figure 4. Additionally, Table 5 shows the notation defined to explain the more detailed operation process of the proposed system shown in Figure 4. In the proposed system, the ODRL, a standardized policy expression language, is employed to transparently and efficiently manage the settlement and distribution of $Copyright_{Royalties}$ and related usage constraints.

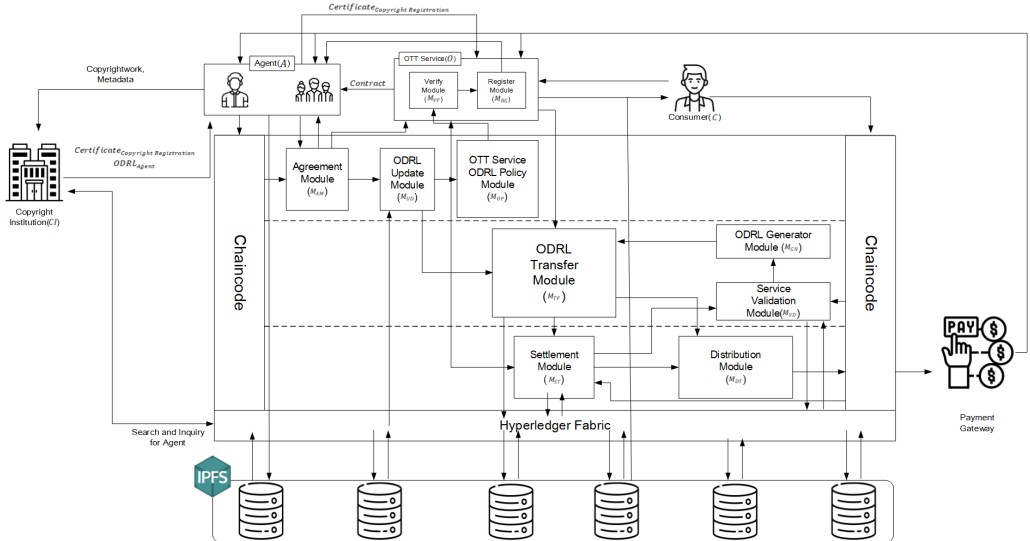

**Figure 4.** Proposed automated OTT service copyright distribution management system using the ODRL.

The operation of the proposed system in this paper is based on the following two assumptions.

(1) It is assumed that when $A$ creates *Content*, $A$ submit metadata, which includes information about $Ratio_{Dis_{Agent}}$ among copyright holders, to $CI$ in each country.

(2) Assuming that $CI$ has received *Content* from the process outlined in Assumption (1), it is assumed that they have verified this *Content*. Once verification is complete, they are presumed to have issued $Certificate_{CopyrightRegister}$ and $ODRL_{Agent}$ based on the received Metadata to $A$.

**Table 5.** Notations.

| Notation | Description |
|---|---|
| $A$ | Agent with multiple copyright holders |
| $O$ | OTT service |
| $C$ | Consumer |
| $C_s$ | Large number of consumers |
| $CI$ | Trusted copyright institution |
| $M_{VR}$ | Verify Module of OTT service |
| $M_{RG}$ | Register Module of OTT service |
| $M_{AM}$ | Agreement module |
| $M_{UD}$ | ODRL update module |
| $M_{OP}$ | OTT service ODRL policy module |
| $M_{TF}$ | ODRL transfer module |
| $M_{ST}$ | Settlement module |
| $M_{GN}$ | ODRL generator module |
| $M_{VD}$ | Service validation module |
| $M_{DT}$ | Distribution module |
| $Contract$ | Copyright transaction related agreement |
| $Content$ | Digital content |
| $Condition$ | Copyright transaction agreement terms and conditions |
| $Certificate_{CopyrightRegister}$ | Copyright registration certificate for digital content |
| $ODRL_{Agent}$ | ODRL for agent |
| $ODRL_{Contract}$ | ODRL for contract |
| $ODRL_{Consumer}$ | ODRL for consumer |
| $Info_{Content}$ | Information about digital content |
| $Info_{OTT}$ | Information about OTT digital content |
| $Info_{Verified}$ | Verified OTT digital content |
| $Info_{Payment}$ | Information regarding payment |
| $Ratio_{Dis_{Agent}}$ | Distribution ratio to agent |
| $Ratio_{Dis_{OTT}}$ | Distribution ratio to OTT service |
| $Stream_{Fee}$ | Amount per stream |
| $Stream_{Nums}$ | Number of streams |
| $Fee_{Consumer}$ | Amount paid by consumer |
| $Rights_{Access}$ | Service access rights |
| $Copyright_{Royalties}$ | Copyright royalties |
| $Contract_{Signed}$ | Signed contracts |
| $Signature_{Shared}$ | Shared signatures |
| $Signature_{Single}$ | Single signature created by aggregating shared signatures |
| $Message_{Register}$ | Message notifying that registration of digital content within the OTT service has been completed |
| $Account_{Consumer}$ | Consumer's account information |
| $Payment_{RatePlan}$ | Payment rate plan |
| $Method_{Pay}$ | Payment method |
| $Info_{Fee_{Consumer}}$ | Information on the amount paid by the consumer |
| $Hash_{Content}$ | Hash value of digital content |
| $Num_{Content_{Play}}$ | Number of plays for digital content |
| $Copyright_{Roy_{Agent}}$ | Copyright royalties to be distributed to agent |
| $Copyright_{Roy_{OTT}}$ | Copyright royalties to be distributed to OTT service |
| $Dis_{Roy_{Agent}}$ | Copyright royalties distributed to agent |
| $Dis_{Roy_{OTT}}$ | Copyright royalties distributed to OTT service |
| $Send(x, y, z)$ | Function of sending $x, y, z$ |
| $Extract(x, y)$ | Function of extracting $y$ from $x$ |
| $Update(x, y)$ | Update function with the addition of $y$ to $x$ |
| $Aggregate(x, y)$ | Function for aggregating $x$ to generate $y$ |
| $Verify(x, y)$ | Function of verifying $x$ by comparing it with $y$ |
| $Divide(x, y)$ | Function of dividing $x$ into $y$ |
| $Generate(x, y, z)$ | Function to generate $z$ based on $x$ and $y$ |
| $Register(x, y)$ | Function of registering $x$ and $y$ |
| $Save(x, y)$ | Function of storing $x$ and $y$ |
| $Lookup(x, y)$ | Function to search $y$ based on $x$ |
| $Return(x)$ | Function that returns $x$ |

**Table 5.** *Cont.*

| Notation | Description |
|---|---|
| $Check(x, y, z)$ | Function of checking $z$ based on $x$ and $y$ |
| $Raise(x, y)$ | Function that increases $x$ based on $y$ |
| $Calculate(x, y, z)$ | Function of calculating $z$ based on the operations of $x$ and $y$ |
| $Distribute(x, y)$ | Function of distributing $x$ and $y$ |

In relation to Assumption (2), once *CI* completes the verification of *Content*, it searches and queries *A*'s information based on the records in Hyperledger Fabric to verify if *A* has been previously registered. If *A* has no prior registration record, a new ODRL code is issued following the same process as earlier. However, if *A* is a digital content production and entertainment company like the 'Walt Disney Company', with a history of *Content* registration, the ODRL information of that *A* is parsed. Subsequently, new metadata related to the *Content* (e.g., $Info_{Content}$, copyright holder information, $Ratio_{Dis_{Agent}}$) is added to the existing $ODRL_{Agent}$. This enables *O* to contract with the digital content production company using a single ODRL, making the contracting process more efficient. The system proposed in this paper operates based on three phases: 'Register', 'Usage', 'Settlement/Distribution', and the key processes of these are as follows.

(1) Register Phase: Through a *contract* with *O*, *A* registers their *Content* on the InterPlanetary File System (IPFS).
(2) Usage Phase: $C_s$ wishing to access *Content* on *O* pay a $Fee_{Consumer}$. This $Fee_{Consumer}$ establishes $C_s$' $Rights_{Access}$ to use the *Content*, resulting in the *Content* being specifically provided to these $C_s$ in alignment with their established $Rights_{Access}$.
(3) Settlement/Distribution Phase: A transparent and fair settlement and distribution of $Copyright_{Royalties}$ are performed based on the $Fee_{Consumer}$ paid by numerous $C_s$.

### 4.1. Register Phase

Figure 5 depicts the procedure of *A* registering their created *Content* on IPFS, facilitated by a contractual agreement with *O*. The specific steps involved in this operation are detailed as follows.

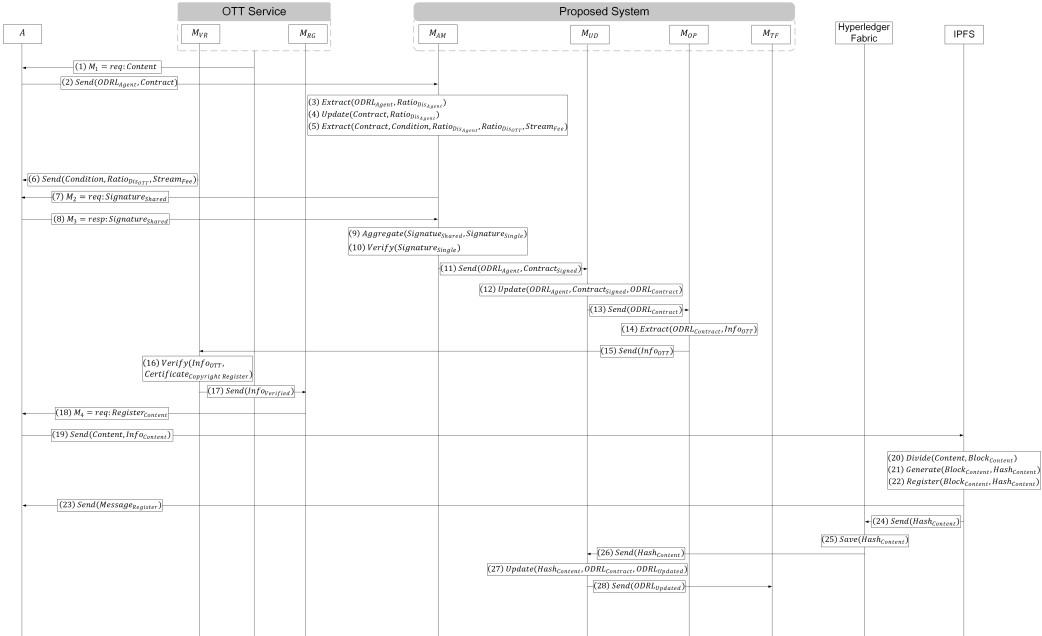

**Figure 5.** The process of the Register phase.

(1) *O* delivers a *Contract* to *A* who owns the copyright for the *Content* to be registered in its service.

(2) *A* delivers the $ODRL_{Agent}$ issued from *CI* to $M_{AM}$ along with the delivered *Contract*.

(3) $M_{AM}$, upon receiving the $ODRL_{Agent}$, extracts $Ratio_{Dis_{Agent}}$ from its contents, applicable when *A* is composed of multiple copyright holders.

(4) $M_{AM}$ proceeds to update the *Contract* by incorporating the $Ratio_{Dis_{Agent}}$ obtained in Step (3).

(5) Following this, $M_{AM}$ extracts the *Condition*, $Ratio_{Dis_{Agent}}$, $Ratio_{Dis_{OTT}}$, and $Stream_{Fee}$ from the updated *Contract* as per Step (4).

(6) $M_{AM}$ sends the extracted *Condition*, $Ratio_{Dis_{Agent}}$, $Ratio_{Dis_{OTT}}$, and $Stream_{Fee}$ to *A* as derived in Step (5).

(7) $M_{AM}$ requests a $Signature_{Shared}$ from *A*.

(8) Copyright holders within *A*, in agreement with the information extracted in Step (5), proceed to sign, thereby sending back a $Signature_{Shared}$ as a response.

(9) $M_{AM}$ aggregates the $Signature_{Shared}$ received in Step (8). Once all copyright holders have signed, a single signature value $Signature_{Single}$ is derived.

(10) $M_{AM}$ then verifies the validity of the $Signature_{Single}$ generated in Step (9).

(11) Upon verifying the $Signature_{Single}$ in Step (10) and confirming the consensus of all copyright holders on the *Contract* terms, $M_{AM}$ sends the $ODRL_{Agent}$ and the $Contract_{Signed}$ to $M_{UD}$. Additionally, *A* forwards the $Certificate_{CopyrightRegister}$, received from *CI*, to *O*'s $M_{VR}$.

(12) $M_{UD}$ creates the $ODRL_{Contract}$ by adding and updating the $Contract_{Signed}$ received in Step (11) to the $ODRL_{Agent}$.

(13) $M_{UD}$ then sends the $ODRL_{Contract}$ to $M_{OP}$.

(14) $M_{OP}$ extracts $Info_{OTT}$ from the $ODRL_{Contract}$, which is used to verify *Condition* in *O*.

(15) $M_{OP}$ sends the extracted $Info_{OTT}$ to *O*'s $M_{VR}$.

(16) $M_{VR}$ compares the information in the $Certificate_{CopyrightRegister}$ received from *A* in Step (11) with the $Info_{OTT}$ received from $M_{OP}$ in Step (15).

(17) $M_{VR}$ of *O*, through Step (16), verifies the match of *Content* information and then sends $Info_{Verified}$ to $M_{RG}$.

(18) Upon receiving $Info_{Verified}$ from $M_{VR}$, which confirms the validation of the *Content* in Step (17), $M_{RG}$ requests $Register_{Content}$ from *A*.

(19) *A*, upon receiving a request for $Register_{Content}$ from $M_{RG}$ within *O*, sends the *Content* and $Info_{Content}$ to IPFS.

(20) IPFS divides the *Content* received from *A* into smaller fragments to create $Block_{Content}$.

(21) A unique $Hash_{Content}$ is generated for the created $Block_{Content}$.

(22) IPFS registers the $Block_{Content}$ created in Step (20) and the $Hash_{Content}$ generated in Step (21) within the IPFS network, thereby forming a distributed file system.

(23) IPFS sends a $Message_{Register}$ to *A*, indicating the completion of the *Content* registration.

(24) IPFS sends the $Hash_{Content}$ to Hyperledger Fabric.

(25) Hyperledger Fabric stores the $Hash_{Content}$ received through process (24) in its ledger.

(26) Hyperledger Fabric then sends the stored $Hash_{Content}$ to $M_{UD}$.

(27) $M_{UD}$ adds the $Hash_{Content}$ received in Step (26) to the $ODRL_{Contract}$, creating an $ODRL_{Updated}$.

(28) $M_{UD}$ forwards the $ODRL_{Updated}$, created in Step (27), to $M_{TF}$.

For the $Info_{Content}$ in the 'Asset' Class of ODRL, a unique identifier such as an International Standard Book Number (ISBN) is used in the uid to distinctly identify the *Content*. Concurrently, for the $Hash_{Content}$ to be stored, as outlined in Step (25), it is stored in 'source' since it denotes the location or origin of the *Content*. The $Hash_{Content}$ is thus added to 'source', indicating the storage location of the *Content*. Following the storage of $Hash_{Content}$, automated *Content* delivery can be executed in the 'Usage' phase, utilizing the $Hash_{Content}$ as requested by *C*.

### 4.2. Usage Phase

Figure 6 explains the 'Usage' phase, where $O$ provides *Content* access to $C_s$ who wish to use the *Content* available on $O$. This process assumes a scenario where a new $C$ who has not paid the $Fee_{Consumer}$ for $O$ wants to use it.

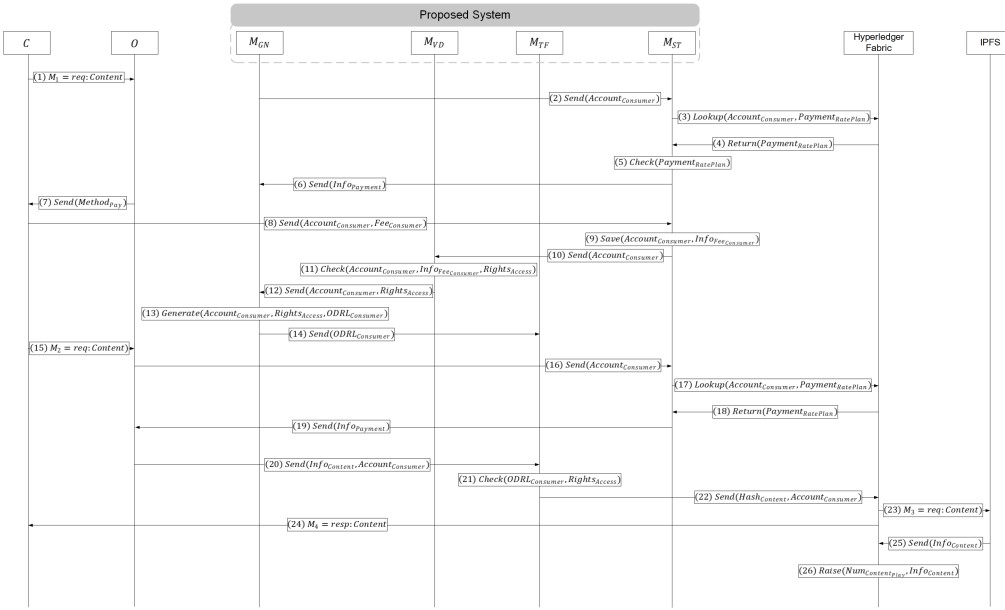

**Figure 6.** The process of the Usage phase.

$C$ desiring to use $O$ creates their $Account_{Consumer}$ by completing the membership registration process. During this, $C$ enters their personal identity information, such as a social security number, into $O$. $O$ then verifies the provided identity information by comparing it with the identification documents. Upon successful verification, $O$ performs a zero-knowledge proof to determine if $C$ is over 19 years old, based on the verified identity information, and stores the generated proof value in the $Account_{Consumer}$ information. Zero-knowledge proof is a cryptographic theory that verifies a fact using only the truth or false of a proposition without revealing any information about the proposition itself [13]. In the proposed system, running in the Hyperledger Fabric environment, using personal information directly in the system without privacy-enhancing technologies can lead to the storage of personal information in a distributed ledger, potentially breaching confidentiality and privacy regarding $C$ identity. To solve this problem, the system employs zero-knowledge proofs to ensure the confidentiality and privacy of $C_s$ by verifying only whether they meet the conditions for using *Content*, without storing personal information on the Blockchain.

(1) $C$ requests playback of the *Content* they wish to use through $O$ that owns the *Content*.

(2) $O$ sends the $Account_{Consumer}$ of $C$ who requested the *Content* playback to $M_{ST}$.

(3) $M_{ST}$ queries the $Payment_{RatePlan}$ based on the $Account_{Consumer}$ in Hyperledger Fabric.

(4) Hyperledger Fabric returns the $Payment_{RatePlan}$ corresponding to the $Account_{Consumer}$ to $M_{ST}$.

(5) $M_{ST}$ checks the $Payment_{RatePlan}$ received in Step (4).

(6) $M_{ST}$, based on the $Payment_{RatePlan}$ checked in Step (5), verifies that $C$ has not made a payment and sends $Info_{Payment}$ to $O$.

(7) $O$, upon confirming that $C$ has not made the required $Fee_{Consumer}$ for using $O$, provides $C$ with limited playback of *Content* along with $Method_{Pay}$.

(8) $C$ sends $Fee_{Consumer}$, corresponding to their desired rate plan, to $M_{ST}$, along with their $Account_{Consumer}$.

(9) $M_{ST}$ stores $Info_{Fee_{Consumer}}$ in the $Account_{Consumer}$.

(10) $M_{ST}$ then sends the updated $Account_{Consumer}$ to $M_{VD}$ as per Step (9).

(11) $M_{VD}$ determines the specific rate plan corresponding to the payment made, as indicated by the $Info_{Fee_{Consumer}}$ in the received $Account_{Consumer}$, and confirms the associated $Rights_{Access}$.

(12) Following this, $M_{VD}$ sends the $Account_{Consumer}$, along with the confirmed $Rights_{Access}$ from Step (11), to $M_{GN}$.

(13) $M_{GN}$ creates $ODRL_{Consumer}$ based on the $Account_{Consumer}$ and $Rights_{Access}$.

(14) $M_{GN}$ then sends the $ODRL_{Consumer}$ to $M_{TF}$.

(15) $C_s$ who have gained $Rights_{Access}$ for the *Content* registered on $O$ re-request the playback of their selected *Content* from $O$.

(16) $O$ sends the $Account_{Consumer}$ details to $M_{ST}$.

(17) $M_{ST}$ retrieves the $Payment_{RatePlan}$ from Hyperledger Fabric, using the $Account_{Consumer}$ information as a reference.

(18) Hyperledger Fabric provides $M_{ST}$ with the $Payment_{RatePlan}$ associated with the $Account_{Consumer}$.

(19) $M_{ST}$ then sends the $Info_{Payment}$ of $C$, who has requested *Content* playback, to $O$.

(20) After verifying that $C$ who requested *Content* playback has completed the payment, $O$ sends both the $Account_{Consumer}$ and the $Info_{Content}$ requested by $C$ to $M_{TF}$.

(21) $M_{TF}$ checks $C$'s $Rights_{Access}$ based on the $ODRL_{Consumer}$ received from $M_{GN}$ during Step (14).

(22) Once $M_{TF}$ confirms that the specific $C$ has gained $Rights_{Access}$ for the *Content*, it sends the $Hash_{Content}$ and $Account_{Consumer}$ details to Hyperledger Fabric.

(23) Hyperledger Fabric, upon receiving the $Hash_{Content}$, instructs IPFS to send the *Content*, which matches the $Hash_{Content}$, to $C$ associated with $Account_{Consumer}$.

(24) Following this request, IPFS delivers the specified *Content* to $C$.

(25) After providing the *Content*, IPFS sends the $Info_{Content}$, detailing the *Content* delivered, back to Hyperledger Fabric.

(26) Hyperledger Fabric then increments the $Num_{Content_{Play}}$ count by one, based on the $Info_{Content}$ it received.

Through the 'Usage' phase, $C_s$ gain the $Rights_{Access}$ for *Content* to access *Content* by paying for $O$. The sequence of actions $C_s$ take to engage with *Content* is documented in Hyperledger Fabric, where the usage history of each *Content* item is also stored. As described in Section 4.3, the 'Settlement/Distribution' Phase aims to protect the rights of copyright holders. This is achieved through a transparent process of settling and distributing $Copyright_{Royalties}$, which is based on the payments made by $C_s$ during the 'Usage' phase and the recorded usage history of the *Content*.

### 4.3. Settlement/Distribution Phase

Through the previously described 'Register' and 'Usage' phases, $O$ collects $Fee_{Consumer}$ from a large number of $C_s$. In the concluding 'Settlement/Distribution' phase, there is a process of transparent and equitable settlement and distribution of the $Copyright_{Royalties}$, which are based on the cumulative usage fees of $O$. Figure 7 illustrates this 'Settlement/Distribution' phase. It demonstrates how transparent settlement and distribution of $Copyright_{Royalties}$ are conducted among various copyright holders, as well as between $A$ comprising multiple copyright holders and $O$. This process is grounded in the $Fee_{Consumer}$ that $C_s$ pay to access $O$. The detailed operational procedure is outlined as follows.

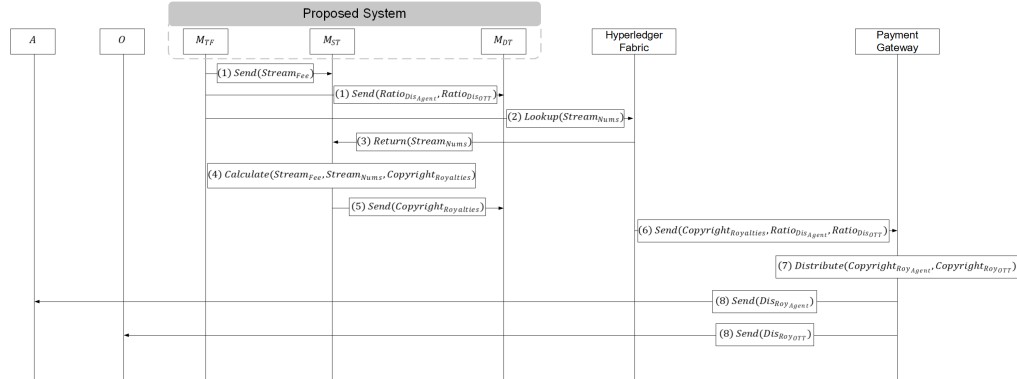

**Figure 7.** The process of the Settlement/Distribution phase.

(1)  In the 'Register' phase as described earlier, $M_{FT}$, after receiving $ODRL_{Updated}$ from $M_{UD}$ in Step (28), extracts $Stream_{Fee}$ and sends it to $M_{ST}$. Additionally, it sends $Ratio_{Dis_{Agent}}$ and $Ratio_{Dis_{OTT}}$ to $M_{DT}$.

(2)  $M_{ST}$ queries Hyperledger Fabric for $Stream_{Nums}$.

(3)  Hyperledger Fabric returns $Stream_{Nums}$ to $M_{ST}$.

(4)  $M_{ST}$ calculates $Copyright_{Royalties}$ by multiplying the $Stream_{Fee}$, received from $M_{FT}$ in Step (1), with the $Stream_{Nums}$ obtained from Hyperledger Fabric in Step (3).

(5)  $M_{ST}$ then sends the calculated $Copyright_{Royalties}$ to $M_{DT}$.

(6)  $M_{DT}$ sends the $Copyright_{Royalties}$, received from $M_{ST}$ in Step (5), along with $Ratio_{Dis_{Agent}}$ and $Ratio_{Dis_{OTT}}$, received from $M_{FT}$ in Step (1), to the Payment Gateway.

(7)  The Payment Gateway, based on the received $Ratio_{Dis_{Agent}}$ and $Ratio_{Dis_{OTT}}$, allocates $Copyright_{Roy_{Agent}}$ to $A$ and $Copyright_{Roy_{OTT}}$ to $O$.

(8)  Finally, the Payment Gateway sends the allocated amounts $Dis_{Roy_{Agent}}$ to $A$ and $Dis_{Roy_{OTT}}$ to $O$, as distributed in Step (7).

Through this process, the automated OTT Service copyright management system utilizing the ODRL can protect the privacy of sensitive data related to *Content* copyrights. By executing *Contract* based on the signatures of all copyright holders, considering multiple stakeholders, the system ensures the protection of authors' rights. Additionally, it automatically verifies $C_s$' $Rights_{Access}$ to provide *Content*. By taking into account joint copyright holders, the system transparently and fairly settles and distributes $Copyright_{Royalties}$. This ensures that copyright holders not only have their rights secured, but also receive fair royalties proportionate to their contributions.

Additionally, many copyright holders currently aim to manage and protect their copyrights effectively through copyright trust management organizations. However, due to inadequate regulations governing the legal relationship between these organizations and the copyright holders, they are often constrained to adhere to the *Contract* established by the trust management organizations. This limitation leads to a challenge for copyright holders in transparently verifying their earnings from these organizations. The proposed system addresses this issue by enabling copyright holders to express their views and enter into *Contract* based on their signatures concerning the $Ratio_{Dis_{Agent}}$ and $Ratio_{Dis_{OTT}}$ of $Copyright_{Royalties}$ proposed by $O$. Furthermore, the system guarantees copyright protection by distributing *Content* according to $C_s$' $Rights_{Access}$ to use this *Content*. Consequently, this approach simplifies the transaction process by eliminating the need for intermediaries like copyright trust management organizations. By reducing agency fees, it ensures that copyright holders receive greater profits and better protection of their rights.

## 5. Main Modules

In the automated OTT service copyright distribution management system utilizing the ODRL, it has been demonstrated that four key goals can be accomplished through the 'Register', 'Usage', and 'Settlement/Distribution' phases. To realize the second goal—ensuring

the distribution of *Content* is based on the collective signatures of all copyright holders, especially in cases involving multiple joint copyright holders — $M_{AM}$ is critical. Additionally, $M_{TF}$ plays a vital role in achieving the third goal of providing *Content* by automatically verifying $C_s$' $Rights_{Access}$ using the information encoded in $ODRL_{Consumer}$. This module also contributes to the fourth goal of efficiently managing the settlement and distribution of $Copyright_{Royalties}$, taking into account joint copyright holders. Therefore, by detailing the operational processes of these two essential modules, the system vividly illustrates their specific functions and roles.

*5.1. Agreement Module*

In scenarios involving multiple copyright holders, $M_{AM}$, responsible for verifying the signatures of all copyright holders, is depicted in Figure 8.

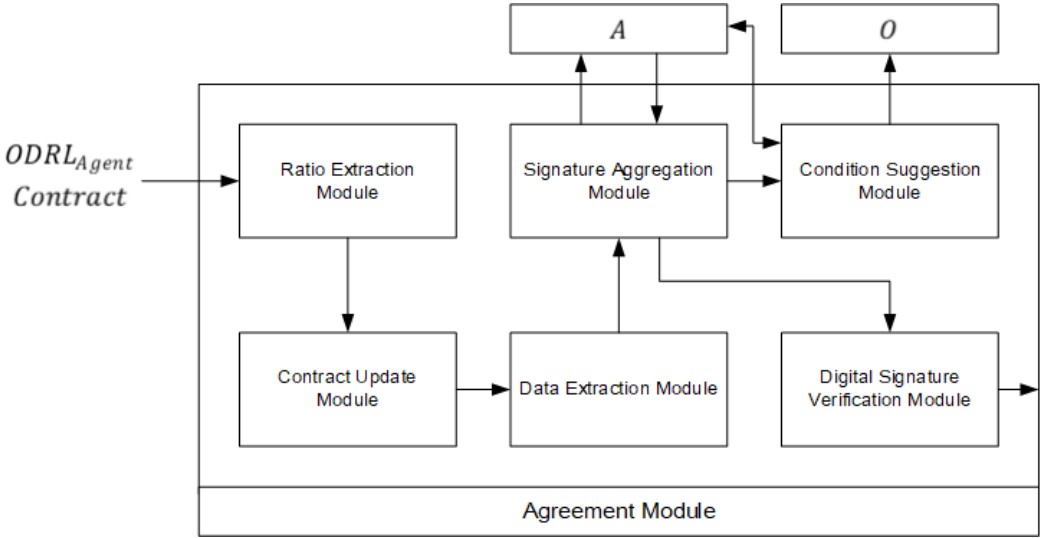

**Figure 8.** Architecture of the Agreement module.

As for the system proposed in this paper, it eliminates the need for an intermediary, such as a trust management organization. Therefore, the term '*A*' is applicable to either an individual or a collective of multiple copyright holders. In situations where *A* comprises several copyright holders, $M_{AM}$ employs the Threshold Signature Scheme (TSS) to ensure the protection of copyrights for all involved copyright holders.

The TSS refers to a technology where a single digital signature is generated when a designated threshold number *t* of participants out of *n*, each perform a partial signature using their private keys. The TSS process is executed as follows [34].

1.  Using a threshold key generation algorithm, a key pair (*P*, *V*) is produced. The set $S=\{S_1, S_2, \cdots, S_n\}$ represents a collection of 'shared secret keys', designed to split the secret key into *n* parts for distribution among multiple participants involved in the signing. The public key, *P*, is utilized for verifying the generated signature, and a verification key, *V*, is also created to validate the shared values of the signature.
2.  Each participant generates a signature share value, $\sigma_i$, using their individual shared secret key and the message *m*.
3.  The signature share value $\sigma_i$ for each participant is verified through *P* and *V*.
4.  The single signature value $\sigma$ is formed by combining a set of *t* verified signature share values $\{\sigma_1, \sigma_2, \cdots, \sigma_t\}$, in accordance with the previously established threshold *t*.
5.  Using the *P*, $\sigma$ is then utilized to validate the signature corresponding to *m*.

Therefore, to ensure that *Contract* is established only when all copyright holders agree on *Contract* terms, the TSS with a threshold value equal to the number of copyright holders *n* is used to guarantee their rights. Typically, in *Content* copyright transactions, the agreement of all joint copyright holders or stakeholders is required. Accordingly, by

considering multiple joint copyright holders, $M_{AM}$, which facilitates copyright transactions based on signatures, aims to align with current copyright transaction trends. For the keys used in the TSS in $M_{AM}$, it is assumed that they have been distributed in advance.

$A$, upon receiving the *Contract* terms from $O$, forwards them to $M_{AM}$ along with their own $ODRL_{Agent}$. If $M_{AM}$ identifies, based on the $ODRL_{Agent}$, that $A$ is composed of multiple copyright holders, it utilizes the Ratio Extraction Module to extract data concerning the $Ratio_{Dis_{Agent}}$. These data are then relayed to the Contract Update Module, which integrates it into the existing *Contract* terms to encompass aspects of $Ratio_{Dis_{Agent}}$. Subsequently, the Data Extraction Module is employed to extract and compile information that will be sent back to $A$. This step is crucial for communicating the *Contract* terms to all copyright holders involved. The specifics of the information to be extracted are outlined in Table 6.

**Table 6.** Information extracted by the Data Extraction Module and passed to $A$.

| Extracted Information | Description |
|---|---|
| $Stream_{Fee}$ | Refers to the amount charged per playback of *Content*, which varies for each piece of digital *Content*. |
| $Ratio_{Dis_{Agent}}$ | Represents the distribution ratio of $Copyright_{Royalties}$ among copyright holders, established for transparent settlement and distribution, considering multiple joint copyright holders. |
| $Ratio_{Dis_{OTT}}$ | Describes the information about the ratio basis for settlements between $A$ and $O$, derived from the $Copyright_{Royalties}$ earned by providing $O$ to multiple $C_s$. |
| $Condition$ | Indicates the $Rights_{Access}$ of $C_s$ of $O$ regarding what actions they can perform with *Content* accessed through the service. These $Rights_{Access}$ may include streaming and storing *Content*. |

The Signature Aggregation Module, upon receiving the information, sends it to all the copyright holders within $A$. Those who agree with the terms of the *Contract* use their 'shared secret key ($S_i$)' to generate their part of the signature, known as the signature share value $\sigma_i$. If they disagree, they abstain from signing. The Signature Aggregation Module then collects all the signature share values received within a predetermined timeframe. If every copyright holder signs, thereby meeting the pre-established threshold, a unified signature value $\sigma$ is formed. This value undergoes verification by the Digital Signature Verification Module. Once the verification process is successfully completed, the Digital Signature Verification Module informs $M_{UD}$ that all copyright holders have agreed to the terms of the *Contract*.

If the single signature value $\sigma$ is not obtained due to some copyright holders refraining from signing, this situation is reported to the Condition Suggestion Module as an inability to fulfill the *Contract*. In this module, all the copyright holders comprising $A$ are queried about proposing new *Contract* terms. If $A$ decides to propose new terms, these are then communicated to $O$. In turn, $O$ presents these new terms to $A$, facilitating a number of *Contract* request and response cycles as initially determined. For instance, if the maximum number of attempts for *Contract* requests and responses is established at five, and if an agreement or new *Contract* terms are not reached within these five cycles, the Condition Suggestion Module informs both $A$ and $O$ that the *Contract* has expired. Consequently, no further requests or responses for the *Contract* will take place.

Under the approach described earlier, a *Contract* concerning *Content* is executed only if all copyright holders concur with its terms. This ensures that copyright holders can ascertain precisely the percentage of royalties they are entitled to receive. Consequently, this facilitates a transparent process for the settlement and distribution of $Copyright_{Royalties}$ among the copyright holders.

### 5.2. ODRL Transfer Module

$M_{TF}$, used for automation by converting $ODRL_{Consumer}$ containing $Copyright_{Royalties}$ settlement, distribution, and restrictions on *Content Rights$_{Access}$* into a format usable in chaincode, can be seen in Figure 9.

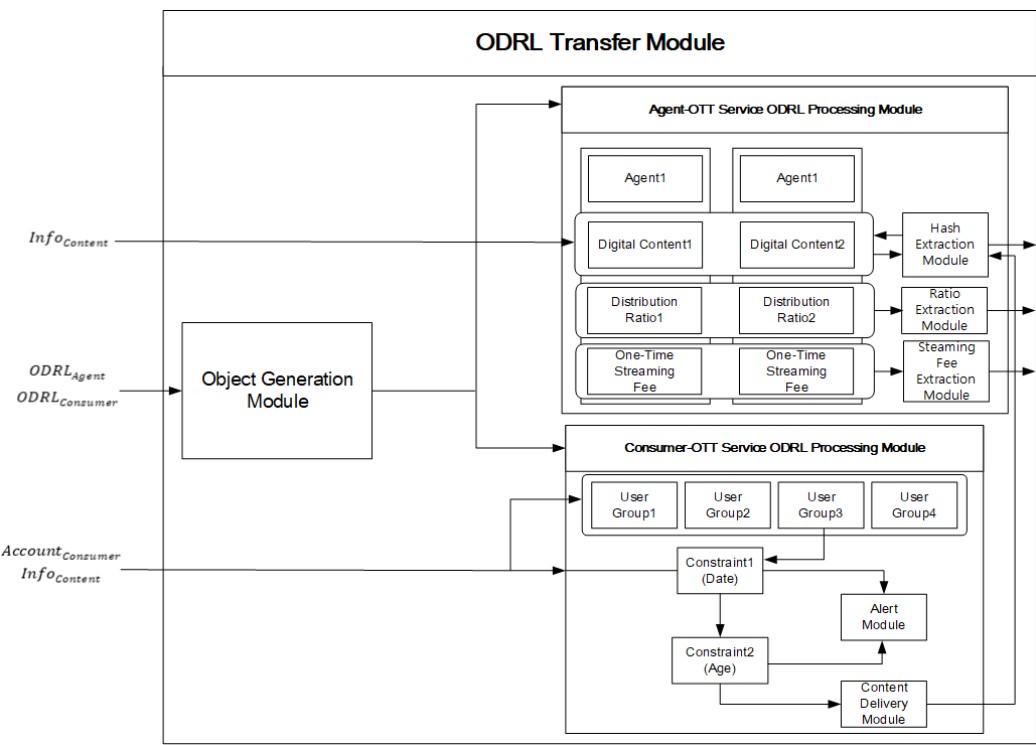

**Figure 9.** Architecture of the ODRL Transfer Module.

In the proposed automated OTT service copyright distribution management system utilizing the ODRL, there exists an $ODRL_{Agent}$ generated based on the contract between *A*, composed of multiple or individual copyright holders, and *O*. Additionally, an $ODRL_{Consumer}$ is also created for the *Contract* between *O* and *C*, based on the amount paid by *C*, outlining *C*'s *Rights$_{Access}$* for the *Content*.

For the $ODRL_{Agent}$ generated between *A* and *O*, it is created based on the *Contract* terms between them. It includes the *Condition*, $Ratio_{Dis_{Agent}}$, $Ratio_{Dis_{OTT}}$, and $Stream_{Fee}$ of *Content* created by *A*. In the case of the $ODRL_{Consumer}$ generated between *O* and *C*, it is the ODRL that includes information about *C*'s *Rights$_{Access}$* to use *Content* based on the selected and paid $Payment_{RatePlan}$, encompassing the specific *Rights$_{Access}$* for each *C*.

In the automated OTT service copyright distribution management system utilizing the ODRL, when these two types of ODRL are generated, they are sent to $M_{TF}$. This process transforms the *content* written in ODRL into a format usable in chaincode, thereby enabling automatic execution of $Copyright_{Royalties}$ distribution between copyright holders, between *A* and *O*, and the control of *C$_s$*' access to *Content*.

Initially, when two ODRLs are introduced into $M_{TF}$, its Object Generation Module undertakes the task of objectifying these ODRLs based on their principal Classes. When adapting the ODRL for use within chaincode, inefficiency and complexity issues may occur. These problems stem from the disparity between the chaincode's operational environment and the varied, intricate rights expressions that ODRL is designed to handle. To address these challenges and ensure uniform processing, objectification is essential. In addition, for this module, objectification is required to determine which 'UserGroup' the *C*, who needs to provide *Content*, belongs to. This determination is based on the *Info$_{Content}$* and $Account_{Consumer}$ provided. Such a process is also essential for efficiently locating the $Hash_{Content}$ of the *Content* being searched for.

When $C$ expresses the desire to access specific *Content* within $O$ and makes a request for it, $O$ transmits the $Info_{Content}$ along with the requesting $C$'s $Account_{Consumer}$ to $M_{TF}$. The Consumer-OTT Service ODRL Processing Module then, utilizing the provided $Account_{Consumer}$ information, identifies the UserGroup $C$ belongs to. A UserGroup is a collective of $C_s$ who share the same expiry date, determined by when they paid for their OTT Service plan. This group is effectively represented by grouping various $C_s$ via the 'PartyCollection', a subset of the 'Party' Class, which is one of the ODRL's main Classes.

The Consumer-OTT Service ODRL Processing Module locates the UserGroup that includes $C$ who requested the *Content* and parses its information. It then compares the constraints listed in the UserGroup information with the received Consumer information and the *Content* to be accessed, to verify if $C$ has $Rights_{Access}$ to use the *Content*.

The first constraint to be verified is whether $C$'s $O$ subscription has expired. If it is confirmed that the expiration date has not passed, the module then checks if $C$'s age meets the age restrictions for the *Content*. This verification is performed using a zero-knowledge proof created when $C$ registered for $O$. For instance, if the *Content* requires $C$ to be over 19 years old, the zero-knowledge proof will yield a true (T) result for $C_s$ above 19 and false (F) for those below. If both the expiry date and age constraint are satisfied, resulting in a T outcome, the Consumer-OTT Service ODRL Processing Module then forwards the $Info_{Content}$ to the Agent-OTT Service ODRL Processing Module. However, if any constraint is not met, the Alert Module is informed about the unfulfilled *Condition*, indicating the requirements needed to access the *Content*.

Once the Consumer-OTT Service ODRL Processing Module confirms that $C$ has met all usage constraints specified in the ODRL, it provides the requested $Info_{Content}$ to the Agent-OTT Service ODRL Processing Module. This module's Hash Extraction Module extracts the $Hash_{Content}$ of the *Content* based on the information received and transmits it to Hyperledger Fabric, enabling the delivery of the corresponding *Content* from IPFS to $C$.

The $ODRL_{Agent}$ created through the contract between $A$ and $O$ contains information about $Copyright_{Royalties}$ settlement and distribution. Based on the received $Info_{Content}$, the Agent-OTT Service ODRL Processing Module conducts a search and retrieval process. Upon finding the desired *Content*, the module sends the $Stream_{Fee}$ of that *Content* to $M_{ST}$, and the extracted $Ratio_{Dis_{Agent}}$, $Ratio_{Dis_{OTT}}$ to $M_{DT}$, ensuring fair settlement and distribution.

Through $M_{TF}$, the $Rights_{Access}$ of $C_s$ for Content are verified, and *Content* is automatically provided to $C_s$ who meet all constraints. Moreover, by automating the processes related to fair settlement and distribution of $Copyright_{Royalties}$ based on the $Stream_{Fee}$ and $Ratio_{Dis_{Agent}}$, $Ratio_{Dis_{OTT}}$ written in the ODRL, efficient and transparent management of *Content* distribution can be achieved.

## 6. Analysis

In this section, we conduct a performance analysis of the consent verification method of the proposed automated OTT service distribution management system using the ODRL. Additionally, we perform a comparative analysis of its functionalities against existing digital content distribution management systems.

### 6.1. Performance Analysis

In this section, we undertake a thorough comparison and analysis of the TSS and conventional digital signature techniques utilized within our proposed system to confirm the agreement of all copyright holders participating in a contract. The signature techniques for comparison are Threshold Schnorr and Schnorr Digital Signatures. We intend to compare and analyze the computational load using the Square-and-Multiply algorithm for both digital signature methods. The automated OTT service copyright distribution management system using the ODRL, as proposed, is designed for managing the distribution of copyrights in digital content like OTT service. This system encompasses numerous copyright holders and users, necessitating the adoption of technologies that mirror the digital content market's dynamics. A key aspect of the system, as discussed in this paper, is its reliance

on a digital signature algorithm to ascertain the consent of copyright holders. Therefore, in this section, we plan to evaluate the overhead, safety, and efficiency by analyzing each digital signature algorithm's performance based on computational load. This analysis takes into account that the proposed system operates on Hyperledger Fabric, a private Blockchain platform.

The Square-and-Multiply algorithm, employed for determining the computational workload, is a method for efficiently computing large numbers. It holds a significant role in cryptography, leveraging the properties of modulo. The execution process of this algorithm, when calculating $x^K \bmod N$, is consistent with the steps described in Algorithm 1 [35].

---

**Algorithm 1** Square-and-Multiply.

---

**Require:** $x \in \{0, \ldots, N-1\}$ and $K = (k_{l-1}, \ldots, k_0)_2$
 1: $r \leftarrow 1$
 2: **for** $i = l - 1$ **downto** $0$ **do**
 3:      $r \leftarrow r^2 \bmod N$
 4:      **if** $k_i = 1$ **then**
 5:          $r \leftarrow r \times x \bmod N$
 6:      **end if**
 7: **end for**

---

In this algorithm, the exponent of the equation to be calculated is first converted into binary. Then, moving from left to right across the exponent's bits, a squaring operation is performed for each bit, and multiplication is carried out when a bit is 1.

Initially, for the Schnorr Digital Signature method, we aim to derive the computational costs needed to obtain the signature value and to perform signature verification using the Square-and-Multiply algorithm. The process for the Schnorr Digital Signature is conducted as follows [36,37].

(1) The Key Authentication Center (KAC) selects two random prime numbers, $q \geq 2^{106}$ and $p \geq 2^{1024}$, and identifies an element $g$ with order $q$.
(2) The user establishes a private key by choosing a random number s from $Z_q$.
(3) The public key $V$ is derived by calculating $V \equiv g^{-s} \bmod p$.
(4) For creating a signature for the message $M$, the signer selects a random number $r$ from $Z_q$ and computes $C \equiv g^r \bmod p$.
(5) Using the one-way hash function $H()$, the value $e = H(M, C)$ is calculated, followed by $y \equiv r + se \bmod q$, using the derived $e$ value.
(6) The derived signature values $(e, y)$ from Step (5) are transmitted to the verifier.
(7) The verifier conducts signature verification by calculating $e \equiv H(M, C \equiv g^y) * V^e \bmod p$ with the received signature values $(e, y)$.

Table 7 presents the computed workload for the Schnorr Digital Signature method, calculated using the Square-and-Multiply algorithm. This calculation is premised on the assumption that $k$ acts as a security parameter relative to the magnitude of $q$, leading to the derived computational requirements for the Schnorr Digital Signature.

**Table 7.** Computational costs for each step in Schnorr Digital Signature.

| Step | Schnorr Digital Signature Computation Volume |
|---|---|
| Process (1) | - |
| Process (2) | - |
| Process (3) | $1.5k - 1$ |
| Process (4) | $1.5k - 1$ |
| Process (5) | 1 |
| Process (6) | - |
| Process (7) | $3k$ |

For Steps (3) and (4) of the Schnorr Digital Signature process, the exponents $s$ and $r$, used in the calculations, belong to $Z_q$ and thus have a bit length set by the bit size $k$ of $q$. Consequently, both steps involve $k - 1$ squaring operations and an average of $\frac{k}{2}$ multiplication operations, resulting in a computational load of $1.5k - 1$ for these steps. In Step (5), only one multiplication is required to calculate $y \equiv r + se \ mod \ q$, and for Step (7), the same cost of computation $3k$ as analyzed in [37] for the verification process of the Threshold Schnorr Digital Signature, is derived.

Next, we delve into the Threshold Schnorr Digital Signature, which is an extension of the Schnorr Digital Signature technique. This signature method involves $n$ signing participants, with a set threshold value of $t$. Each participant is denoted as $v_i$ for $i = 1, \cdots, n$. The parameters $p$, $q$, and $g$ are chosen and established under the same criteria as the previously described Schnorr technique. The specific steps involved in these processes are as follows [37].

(1) The signing participant selects a random number $s$ from $Z_q$ as their private key and derives the public key $V$ by calculating $V \equiv g^{-s} \ mod \ p$.

(2) A trustworthy dealer chooses a secret polynomial $A(z) = a_0 + a_1 z + a_2 z^2 + \cdots + a_{t-1} z^{t-1} \ mod \ q$ and distributes $(x_{v_i}, A(x_{v_i}))$ to each participant $v_i$. Here, $x_{v_i}$ is a public value, while $A(x_{v_i})$, calculated based on it, is a secret value.

(3) Participant $v_i$ computes the secret value $x_{v_i} = l_{v_i} \cdot A(x_{v_i}) \ mod \ q$. $l_{v_i}$ is the Lagrange coefficient $\Pi_{1 \le k \le t, k \ne i} \left( \frac{x_{v_k}}{x_{v_k} - x_{v_i}} \right) \ mod \ q$, leading to $s = \Sigma_{v_i \in P} s_{v_i} \ mod \ q$.

(4) The dealer selects a secret polynomial $B(z) = b_0 + b_1 z + b_2 z^2 + \cdots + b_{t-1} z^{t-1} \ mod \ q$, satisfying $b_i \in Z_q^*$ and $b_0 = r$, and then distributes $(x_{v_i}, B(x_{v_i}))$ to each participant $v_i$. Here, $r$ is a secret entrusted value meeting $r \in Z_q^*$. $x_{v_i}$ is a public value, and the computed $B(x_{v_i})$ is a secret value sent to $v_i$.

(5) Signing participant $v_i$ calculates the secret value $r_{v_i} = l_j \cdot B(x_{v_i}) \ mod \ q$, and the total trust value $r$ is then computed as $r = \Sigma_{v_i \in P} r_{v_i} \ mod \ q$.

(6) The signature aggregator broadcasts the message $M$ to be signed to all signing participants. Any participant in the signature process can assume the role of the signature aggregator.

(7) Each signing participant computes their partial commitment value, $c_{v_i} = g^{r_{v_i}} \ mod \ p$, and then broadcasts it.

(8) Signatory participant $v_i$ selects $t$ values from $c_{v_i}$, calculates $e \equiv H(M, \Pi_{v_k \in P} c_{v_k} mod p)$ and $y_{v_i} \equiv r_{v_i} + s_{v_i} e \ mod \ q$, and sends the resulting pair $(e, y_{v_i})$ to the signature aggregator.

(9) The signature aggregator compiles the final signature value $(y, e)$ by processing the $t$ partial signatures $(e, y_{v_i})$ with the same $e$ value.

(10) The verifier conducts signature verification using the received signature value $(e, y)$, following the same method as the Schnorr Digital Signature, by calculating $H(M, C \equiv g^y) * V^e \ mod \ p$.

The Threshold Schnorr Digital Signature employs the same Square-and-Multiply algorithm as the Schnorr Digital Signature previously mentioned. By considering $k$ as a security parameter in relation to the size of $q$, the calculation required for this specific signature method is determined. The computational details for each step of this signature process are outlined in Table 8.

In this paper, the proposed automated OTT service copyright distribution management system using the ODRL stipulates that copyright distribution transactions proceed only when all copyright holders have signed. Consequently, for the computational requirements of generating and verifying Threshold Schnorr Digital Signature, the threshold has been set equal to the number of signatory participants, denoted as $n$.

**Table 8.** Computational costs for each step in Threshold Schnorr Digital Signature.

| Step | Threshold Schnorr Digital Signature Computation Volume |
|---|---|
| Process (1) | $(1.5k - 1)n$ |
| Process (2) | $(n - 1)n$ |
| Process (3) | $(32n - 3)n$ |
| Process (4) | $(n - 1)n$ |
| Process (5) | $(32n - 3)n$ |
| Process (6) | - |
| Process (7) | $(1.5k - 1)n$ |
| Process (8) | $n$ |
| Process (9) | 0 |
| Process (10) | $3k$ |

In Steps (1) and (7), D.H. Nyang's work [37] was found to have errors in the computational estimations for signature generation using the Square-and-Multiply algorithm. These errors were corrected, leading to accurate recalculations of the computational load for these steps. The values for Steps (2), (3), (4), (5), and (10) were based on the 'Schnorr Digital Signature-based Threshold Signature Technique' analysis, as proposed by D.H. Nyang in [37]. In Step (8), there are a total of $n$ multiplications executed, comprising of $n - 1$ multiplications to calculate $e \equiv H(M, \Pi_{v_k \in P} c_{v_k} mod p)$ and an additional single multiplication for $y_{v_i} \equiv r_{v_i} + s_{v_i} e \ mod \ q$. For Step (9), which involves combining partial signatures into a single signature, only addition is used, resulting in a multiplication workload of zero in the Square-and-Multiply algorithm.

Building upon the previously derived computational loads for each step of the Schnorr Digital Signature and Threshold Schnorr Digital Signature, this study aims to compare and analyze the computational demands of these two digital signature methods. This comparison and analysis will focus on the aspects of signature generation and the verification of the resultant signature values. In terms of signature generation, unlike the Schnorr Digital Signature where each participant independently derives a signature value using their private and public keys, the Threshold Schnorr Digital Signature process involves each participant combining shared signature values, derived using their private and public keys, to produce a single signature value. Derived from the data in Tables 7 and 8, the computed costs for signature generation and verification in the Schnorr Digital Signature and Threshold Schnorr Digital Signature are presented in Table 9.

**Table 9.** Computation costs of Schnorr Digital Signature and Threshold Schnorr Digital Signature.

| | Schnorr Digital Signature | Threshold Schnorr Digital Signature Used in the Proposed System |
|---|---|---|
| Signature Generation | $(3k - 1)n$ | $66n^2 + 3kn - 9n$ |
| Signature Verification | $3kn$ | $3k$ |

For the Schnorr Digital Signature, the computational costs per process as shown in Table 7 are based on a scenario with a single signer. To facilitate a clear comparison with the Threshold Schnorr Digital Signature, it is essential to adjust this to a scenario with $n$ signatory participants. Thus, the computational costs listed in Table 9 were derived by multiplying the costs from Table 7 by $n$.

Initially, based on Table 9, we aim to compare and analyze both digital signature techniques from the perspective of signature generation, with $n$ signing participants. It is evident that the Threshold Schnorr Digital Signature, with the threshold set to $n$, the same as the number of signing participants, requires more computational load compared to the Schnorr Digital Signature. This is analyzed to be due to the need to derive shared signature values using secret values $s_{v_i}$ and entrusted values $r_{v_i}$ for the use of the TSS, unlike typical digital signatures. Moreover, the process of combining the derived shared signatures to produce a single key is carried out through addition, which is not included in the computational load derived through the Square-and-Multiply algorithm. Therefore,

when considering these additional calculations, it is discerned that a higher computational load is required.

From the perspective of signature verification, it is noted that the Threshold Schnorr Digital Signature requires *n* times less computational load than the Schnorr Digital Signature. This is attributed to the fact that while the Schnorr Digital Signature requires the verification of each participant's signature value, the Threshold Schnorr Digital Signature performs verification on a single, combined signature value derived from the shared signatures.

In the system proposed in this paper, which is based on Threshold Schnorr Digital Signature, contracts are processed by obtaining signatures from copyright holders within the Agent. While this may lead to some overhead in signature generation compared to traditional Schnorr Digital Signatures, it shows efficiency in processing time during signature verification. Notably, with multiple copyright holders within an Agent, the system's verification time becomes more efficient by a factor of *n*, especially as the number of copyright holders increases.

However, in the system proposed in this paper, since both signature generation and verification processes are implemented, we aim to compare the total computational workload of the two signature techniques, Schnorr Digital Signature and Threshold Schnorr Digital Signature. The total computational requirement needed for these two signature techniques is calculated by adding the costs required for signature generation and verification, as outlined in Table 9 for each technique. Accordingly, the Schnorr Digital Signature requires a computational workload of $6kn - n$, while the Threshold Schnorr Digital Signature requires $66n^2 + 3kn - 9n + 3k$. This relationship can be seen in Figure 10, where the *x*-axis represents the bit size of the private key, the *y*-axis represents the number of signatories, and the *z*-axis signifies the computational requirement.

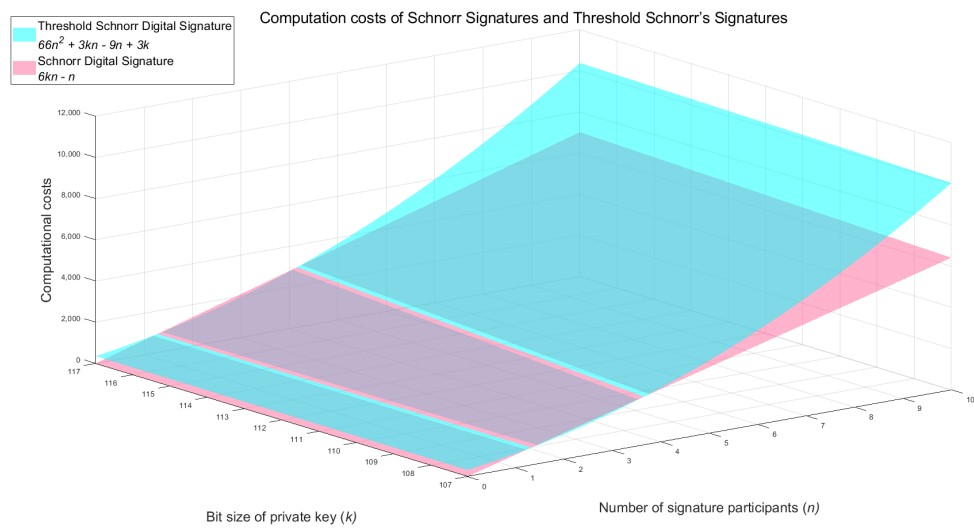

**Figure 10.** Total computational costs graph of Schnorr Digital Signature and Threshold Schnorr Digital Signature.

In the formulas representing the total computational requirements for the two signature techniques, the variable *n* signifies the number of signing participants, and *k* denotes a security parameter related to the size of *q*. Considering the assumption that the prime number *q* is at least $2^{106}$, it is inferred that *k* is 107 or more. Since both *n* and *k* are variables in the formulas that quantify the total computational requirements of these signature methods, to determine the optimal state of the signature technique used in this system, a graph was created in a three-dimensional space. This graph illustrates the total computational requirements, taking into account both variables.

Upon analyzing the two graphs, it has been confirmed that when *n* exceeds 1 but remains below 4, the computational workload for the Threshold Schnorr Digital Signature,

as utilized in the system proposed in this paper, is lower than that of the Schnorr Digital Signature. Consequently, when assessing whether all authors agree to the contract terms using the Threshold Schnorr Digital Signature, particularly in the case of an agent composed of two or three authors, it has been observed that all authors can be identified more quickly than when employing the Schnorr Digital Signature. This discovery implies the potential for establishing a more efficient digital content distribution and management system by obtaining consent to the contract terms from all parties involved.

Furthermore, in terms of system security, traditional Schnorr Digital Signatures require regenerating and reverifying the signature if a private key is lost during verification. However, with the Threshold Schnorr technique, the policy allows contracts to proceed if $t$ out of $n$ copyright holders sign. For example, in a situation where five copyright holders have interests in a single piece of content, a contract can be executed with the signatures of three copyright holders, as per system policy. This ensures stable system operation even if two holders lose their shared key value, as the contract can still be executed with the remaining three signatories.

However, in the context of the system proposed in this paper, there is no advantage in terms of system stability because transactions are executed solely upon the signed consent of all copyright holders associated with each digital content, following legal requirements for copyright transactions. Nevertheless, there is a time efficiency advantage in the case of digital content transaction contracts involving agents consisting of two or three copyright holders who establish interests in specific digital content. This advantage stems from the swift signature and verification speeds facilitated by the characteristics of digital signatures.

## 6.2. Functional Analysis

In this section, we analyze the automated OTT service distribution management system using the ODRL proposed in this paper by comparing it with the functions of the system proposed in the previous study analyzed in Section 3. This analysis can be seen in Table 10. Table 10, the symbol 'O' indicates that the system proposed in the paper provides the respective functionality, 'X' denotes the absence of that functionality, and '△' signifies that only a part of the functionality is offered.

**Table 10.** Comparative analysis of functions between previous study and the proposed system.

| Ref | Copyright Distribution Management Function | Smart Contract Signing Function | Royalty Settlement and Distribution Function | Privacy Protection Function | User Permission Check Function |
|---|---|---|---|---|---|
| [1] | O | X | X | X | X |
| [27] | O | X | O | X | X |
| [28] | O | X | O | X | X |
| [29] | O | X | X | X | X |
| [30] | O | X | X | △ | X |
| [31] | O | X | O | X | X |
| [4] | O | X | O | X | X |
| [3] | O | X | X | X | X |
| Proposed System | O | O | O | O | O |

As analyzed in Section 3, the systems proposed in most studies provide major functions for the purpose of copyright distribution management, but there was a limitation in that they did not design features like 'smart contract signing function', 'copyright royalties settlement and distribution function', 'privacy protection function', and 'user permission check function'.

Accordingly, this paper proposed the automated OTT service distribution management system using the ODRL to solve these limitations and construct a system that can be practically applied to the digital content copyright trading market. Looking at the ecosystem of the current digital content copyright distribution management market, most

copyright holders delegate their rights in copyright transactions through a trust management organization, so most copyright holders cannot check the copyright royalties for their works and the copyright royalty settlement and distribution ratio. Accordingly, the system proposed in this paper is based on chaincode in the Hyperledger Fabric environment to create an automated and active copyright distribution management system, and a number of people who have interests in each digital content are established. In order to guarantee the rights of joint copyright holders, a system was created to directly participate in the distribution of OTT services by signing, so that rights can be guaranteed. In addition, by calculating the copyright royalty based on the number of streams, all copyright holders can transparently check the copyright royalties, and by specifying the copyright royalties settlement and distribution ratio within ODRL, the copyright royalties are settled and distributed transparently and fairly.

Additionally, most systems proposed in previous studies do not consider features for users or consumers. Recognizing that not only creative copyright holders, but also users and consumers play a pivotal role in the vitality of the digital content market and OTT services, the system proposed in this paper includes functionalities catered to users and consumers. Typically, users or consumers provide their identity information for verification before registering to participate in OTT services and digital content copyright trading markets. However, a common concern is the provision of unnecessary personal information and the potential for privacy infringement if personal information is leaked by the service provider. In response, the proposed system utilizes zero-knowledge proof technology to minimize the direct exposure of personal information, thereby enhancing privacy protection. Furthermore, users and consumers, who pay a certain amount for OTT services, should be entitled to corresponding rights. Hence, the proposed system ensures that users and consumers receive services commensurate with the amount paid, based on a user rights verification feature built upon chaincode. This process actively follows the conditions specified within ODRL.

The system proposed in this paper significantly differs from most existing studies in its effort to safeguard the rights of joint copyright holders and to ensure rightful privileges for consumers and users. Additionally, it presents privacy protection as a major feature and advantage, aligning with the growing importance of personal information protection in contemporary contexts.

## 7. Discussion

The paper discusses how, in the OTT service market, illegal replication and distribution of digital video content are rampant due to issues like stream ripping, indiscriminate password sharing among users, credential fraud, fraudulent consumer endpoints, and CDN man-in-the-middle attacks, continuously infringing upon the rights that copyright holders should be guaranteed. Additionally, it confirms the 'black box' problem where, when copyright holders partially entrust the management of their digital content rights to agents like trust management organizations, there is a lack of transparency in tracking the usage history of digital content and in the settlement and distribution of copyright royalties.

To address these limitations, this paper proposes the Automated OTT Service Copyright Distribution Management System Using the ODRL, aimed at preventing the infringement of copyright holders' rights arising from the increasing demand and diversity of digital content, and ensuring transparent and fair copyright royalty settlement and distribution.

The system presented in this paper automatically validates whether a consumer is eligible to use digital content, based on the restrictions defined in ODRL. It allows only authorized users to have limited access to the digital content. Additionally, considering the inherent characteristics of digital content and the presence of multiple co-copyright holders with vested interests, the system employs Threshold Schnorr Digital Signature to safeguard the rights of all copyright holders. This ensures that copyright transactions proceed only when there is unanimous agreement on the transaction terms among all holders. This method enables copyright holders to transparently monitor their share in

copyright royalties, thereby ensuring their rights and facilitating a fair and transparent process for the settlement and distribution of these royalties.

However, besides the problems with unclear settlement and distribution of copyright royalties, there have recently been significant challenges related to joint copyright holders in digital content copyright transactions, resulting in legal disputes [38–40].

A joint work is recognized only if it meets two essential requirements. Firstly, the work must not allow for individual usage; this means joint copyright holders should exercise their intellectual property rights without violating the moral rights of their counterparts. Essentially, exercising copyright rights requires the unanimous agreement of all involved co-copyright holders. Secondly, the creation of such a work requires the collaborative effort of two or more individuals. Reflecting the nature of joint works, it is not possible for a single person to hold the copyright alone; it must involve at least two individuals, all of whom should have the clear intent to be joint copyright holders from the onset of the creation process.

However, the recent increase in legal disputes arising from joint copyright holders not adhering to such legal requirements has consistently led to calls for measures to ensure their rights. Yet, these discussions have mostly been confined to legal solutions.

The system in question leverages TSS technology to ensure the comprehensive protection of rights for joint copyright holders involved in particular digital content. This setup allows copyright transactions to proceed only with the unanimous consent of all copyright holders. This approach enables copyright holders to transparently monitor their share in the distribution of copyright royalties. Furthermore, the system automates this process, ensuring not just the protection of co-copyright holders' rights but also the transparent and equitable settlement and distribution of copyright royalties. This innovation addresses and potentially resolves prevalent issues in the current OTT service market.

As analyzed in Section 6, the proposed system employing Threshold Schnorr Digital Signature was found to be more efficient in digital content distribution management when an agent composed of only two or three copyright holders participates. Specifically, it was observed that in such scenarios, agreement on contract terms from all copyright holders could be obtained in a shorter time compared to using the Schnorr Digital Signature. However, with digital content like movies or dramas on OTT services, the complexity of digital content production requires the involvement of various stakeholders, such as directors, writers, actors, and production companies. As a result, since most digital content cannot be created with only two or three copyright holders, there exists a limitation in efficiently managing transparent copyright distribution for digital content through the system using TSS.

Moreover, the process of conducting copyright transactions and contracts for digital content involves the transmission of sensitive data on the system. The exposure of such data could significantly raise the likelihood of disputes, either between companies or between companies and individuals. To mitigate this risk, the proposed system employs zero-knowledge proof technology, like zk-SNARKs and ZEC, to safeguard individual-identifying sensitive data, aiming to avert privacy infringement issues. However, while the necessity of a Trusted Third Party (TTP) during the initial setup for enhancing system efficiency is typically not a concern in general systems, it presents a unique challenge for the proposed system. This system is based on the private Blockchain platform Hyperledger Fabric, where employing a TTP during the initial setup could potentially compromise the Blockchain's inherent feature of decentralization.

## 8. Conclusions

This paper introduces an automated OTT service copyright management system, leveraging ODRL and chaincode technology. The system is adept at automatically authenticating consumers' rights to access digital content, while simultaneously considering the interests of joint copyright holders. It facilitates the transparent settlement and distribution of royalties, with a goal to protect the rights of copyright owners. When applied to the

current OTT services and digital content markets, this system is anticipated to stimulate the digital content trade market and foster a trustworthy and reliable OTT service environment, enhancing its overall credibility.

Additionally, the proposed system, by utilizing TSS technology, ensures that the exercise of copyright-related transactions and contracts, such as intellectual property rights, is permitted only when all copyright holders agree to the terms. This is a significant advantage. However, as the number of joint copyright holders increases, the system encounters limitations in terms of efficiency. Furthermore, recognizing the potential for legal disputes due to sensitive data in digital content copyright transactions, the system employs zero-knowledge proof technology to reduce the likelihood of privacy breaches. Nonetheless, the reliance on trusted third parties in zero-knowledge proofs could potentially compromise the decentralization of the system's foundational Hyperledger Fabric, presenting a challenge.

Due to the prevalence of multiple joint copyright holders in much of the digital content within the OTT service market, the currently proposed system encounters practical implementation challenges. To overcome these, future developments will aim to incorporate privacy enhancing technologies, which minimize privacy infringement risks while preserving the decentralized features of Blockchain technology. Furthermore, the system will be tailored to efficiently and logically ascertain the consent of numerous joint copyright holders. This strategy is intended to construct an OTT service copyright distribution management system that is feasible for real-world application in the OTT service market.

**Author Contributions:** W.S. designed the proposed system with Performance analysis and drafted this article. S.K. reviewed the article and provided inputs on functional analysis. S.O. revised this article. J.-H.L. revised this article and supervised all the work. All authors have read and agreed to the published version of the manuscript.

**Funding:** This work is supported by the Ministry of Culture, Sports and Tourism and Korea Creative Content Agency (Project Number: CR202104006) and by hte Institute of Information and communications Technology Planning and Evaluation (IITP) under the Metaverse Support Program to Nurture the Best Talents (IITP-2023-RS-2023-00254529) grant funded by the Korean government (MSIT).

**Data Availability Statement:** Data are contained within the article.

**Conflicts of Interest:** Author SungHeun Oh was employed by the company DigiCAP Co., Ltd. The remaining authors declare that the research was conducted in the absence of any commercial or financial relationships that could be construed as a potential conflict of interest.

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
