# Peer review of "Automated Over-the-Top Service Copyright Distribution Management System Using the Open Digital Rights Language"

_electronics, doi:10.3390/electronics13020336_

Round 1
Reviewer 1 Report
Comments and Suggestions for Authors
In the paper, the authors proposed an automated Over-The-Top (OTT) service copyright distribution management system using the Open Digital Rights Language (ODRL) to address the infringement of copyright holders’ rights arising from the increased demand and diversity in digital content. The performance of the proposed approach has been highlighted based on a comparative analysis with the conventional Schnorr digital signature, in terms of computational effort for signature generation and verification. However, there are some aspects which the authors should consider to improve the structure and quality of the paper:
- Please remove the abbreviations „OTT” and „ODRL” from the title of the paper, replacing them with the full forms.
- In the Abstract section, please remove any abbreviations. Please include a special paragraph in the main body of the article.
- In the introduction, the authors did not include any references. To highlight the importance of the proposed topic, the authors should demonstrate that it is worthy of investigation starting from studies carried out in recent years.
- Paragraph 2.1 is based on only two references. The number is small compared with what has been studied and published recently regarding smart contracts. The same observation is done for Paragraph 2.2 (one reference). Please redo the state-of-the-art for these paragraphs.
- Please specify the meaning of the symbols from Tables 3 and 9.
- The quality of Figures 3 and 4 should be improved. The figures are unclear.
- The authors asserted in paragraph 6.1. “This analysis takes into account that the proposed system operates on Hyperledger Fabric, a private blockchain platform.” Please introduce the reference for the Hyperledger Fabric platform.
- The limits of the proposed approach should be better highlighted.
- The reference list should be extended with more references, as I pointed out in points 3 and 4.
Author Response
Thanks for your comments. Plz find the attached file.

Reviewer 2 Report
Comments and Suggestions for Authors
The authors propose a novel technique of copyright distribution based on blockchain and smart contracts.
The advantages of the article are as follows:
1) A new language (ODRL) for smart contracts is proposed based on the Hyperledger platform.
2) The authors describe basic templates for copyright distribution and royalty distribution among multiple copyright holders.
3) A detailed structure of blockchain blocks and control flow is given.
4) A good literature review is given.
Disadvantages of the text:
1) The authors need to report initial tests of the usage of the proposed solution. Does this solution only exist on paper? I don't think so.
2) Blockchain-based solutions are very reliable, but frauds are not eliminated. The authors do not report any cybersecurity analysis of the solution.
3) Some text in Fig. 4 is so tiny that it is unreadable.
Good.
Author Response

(The authors gave the same response as above.)

Reviewer 3 Report
Comments and Suggestions for Authors
This paper proposes an automated OTT service copyright distribution management system using ODRL to secure copyright holders' rights. The system facilitates joint copyright holders' participation in OTT service distribution, ensuring fair royalty distribution and addressing privacy concerns through zero-knowledge proof technology. The approach aims to create a protected OTT service market, resolving issues of illegal content copying and unfair royalty settlements.
1) Introduction:
a. Could the introduction provide more context on the specific challenges copyright holders face in the digital content landscape, aside from illegal copying and distribution?
b. Can the paper elaborate on the significance of resolving these challenges for the broader digital content industry?
2) Problem Statement:
a. Could the paper delve deeper into the potential consequences of the identified challenges on the digital content industry?
3) Solution Alignment:
a. How does the proposed system using ODRL align with or differ from existing solutions to copyright-related issues in OTT services?
b. Are there alternative technologies or approaches considered in the literature that could be compared with the proposed system?
4) Handling Multiple Copyright Holders:
a. Can the paper provide more insights into scenarios where joint copyright holders may have conflicting interests and how the proposed system addresses such conflicts?
b. What are the computational overheads associated with implementing the TSS mechanism, especially when dealing with a large number of copyright holders?
5) Privacy Protection Measures:
a. Could the paper provide specific details on the privacy protection measures employed, especially in handling sensitive data during copyright distribution and royalty settlement?
b. Are there any potential vulnerabilities or limitations in the privacy protection measures that should be acknowledged and addressed?
6) Comparative Analysis:
a. Can the paper elaborate on the specific scenarios or use cases where the proposed system's use of Threshold Schnorr Signatures outperforms the conventional Schnorr digital signature?
b. Are there any potential trade-offs or limitations associated with the efficiency gains achieved by the proposed system?
7) Future Directions:
a. What are the specific challenges or research gaps that the paper envisions addressing in future research to enhance the proposed system?
b. How might external factors, such as evolving technologies or industry trends, impact the feasibility and success of the proposed future directions?
8) Technical Concepts Clarification:
a. Could the paper provide brief definitions or references for technical concepts like smart contracts and ODRL to assist readers unfamiliar with these terms?
9) Grammar and Style:
a. Could the paper undergo a final proofread to ensure consistency in grammar and style throughout?
Comments on the Quality of English LanguageModerate editing of English language required
Author Response

(The authors gave the same response as above.)

Round 2
Reviewer 1 Report
Comments and Suggestions for Authors
The authors performed changes to the initial manuscript. New explanations, elaborations of details, and revisions have been added. I no longer have any observations.
Author Response
Thanks for your agreement on the revised paper! Thanks again for your previous comments that made this paper significantly improved.
Reviewer 2 Report
Comments and Suggestions for Authors
The authors adressed properly all my doubts.